# A multiple charging correction algorithm for broad supersaturation scanning cloud condensation nuclei (BS2-CCN) system

Najin Kim[1,2,a], Hang Su[1], Nan Ma[3], Ulrich Pöschl[1], Yafang Cheng[2]

[1]Multiphase Chemistry Department, Max Planck Institute for Chemistry, Mainz, 55128, Germany
[2]Minerva Research Group, Max Planck Institute for Chemistry, Mainz, 55128, Germany
[3]Center for Air Pollution and Climate Change Research (APCC), Institute for Environmental and Climate Research (ECI), Jinan University, Guangzhou, 511443, China
[a] Currently at: Center for Sustainable Environment Research, Korea Institute of Science and Technology

*Correspondence to*: Yafang Cheng (yafang.cheng@mpic.de)

**Abstract.** High time resolution (~1s) of aerosol hygroscopicity and CCN activity can be obtained with a Broad Supersaturation Scanning Cloud Condensation Nuclei (BS2-CCN) system. Based on a commercial DMT-CCNC, the newly designed diffusive inlet in the BS2-CCN realizes a broad supersaturation distribution in a chamber with a stable low sheath to aerosol flow ratio (SARs). In this way, a monotonic relation between activation fraction of aerosols ($F_{act}$) and critical activation supersaturation ($S_{aerosol}$) can be obtained. The accuracy of the size-resolved aerosol hygroscopicity, κ, measured by the BS2-CCN system can

be, however, hampered by multiply charged particles, i.e., resulting in the overestimation of κ values. As the BS2-CCN system uses multiple and continuous supersaturations in the chamber and the size-resolved $F_{act}$ value is directly used to derive κ values, the multiple charging correction algorithm of the traditional CCNC where single supersaturation is applied does not work for the BS2-CCN observation. Here, we propose a new multiple charging correction algorithm to retrieve the true $F_{act}$ value. Starting from the largest size bin, a new $F_{act}$ value at a specific particle diameter ($D_p$) is updated from a measured

activation spectra after removing both aerosol and CCN number concentration of multiply charged particles using a Kernel function with a given particle number size distribution. We compare the corrected activation spectra with laboratory aerosols for a calibration experiment and ambient aerosols during the 2021 Yellow-Sea Air Quality Studies (YES-AQ) campaign. It is noted that this algorithm is only applied to the monomodal particle distribution. The difference between corrected and measured κ values can be as large as 0.08 within the measured κ values between 0.11 and 0.37 among the selected samples,

highlighting that multiple charge effect should be considered for the ambient aerosol measurement. Furthermore, we examine how particle number size distribution is linked to the deviation of activation spectra and κ values.

## 1 Introduction

Cloud condensation nuclei (CCN), the subset of atmospheric aerosol particles that can activate at certain supersaturation to form cloud droplets, are a key element in global climate change as they modulate the microphysical properties of the clouds
(e.g., number concentration, mass and effective radius). Despite the scientific importance of CCN, there are still large

uncertainties in assessing the aerosol-cloud interactions and quantification of their effect on climate due to the complexity of atmospheric composition and processes (IPCC 2007). To reduce these uncertainties, knowledge of the spatio-temporal distribution of CCN and their activation characteristics is essential. As part of these efforts, numerous field campaigns have been carried out in the various regions in recent years (Andrae and Rosenfeld 2008; Chang et al., 2009; Che et al., 2016; Ervens et al., 2009; Gunthe et al., 2009; Hämeri et al., 2001; Hudson 1993; Jurányi et al., 2011; Kim et al., 2014; Moore et al., 2012; Ovadnevaite et al., 2011; Pöhlker et al., 2016; Rose et al., 2010; Schmale et al. 2018; Su et al., 2010; Thalman et al., 2017; Xu et al., 2021 and reference therein).

The activation of CCN at a given level of supersaturation can be primarily determined by particle size, followed by chemical composition and mixing state (Dusek et al., 2006; Ervens et al., 2009; Ren et al., 2018; Padró et al., 2012). Therefore, the size-resolved CCN measurement can separate the size effect to investigate the chemical composition effect on CCN activation efficiency under the simple assumption. This can help improve the understanding of CCN activation characteristics. Also, the critical diameter at a given supersaturation can be determined. For the size-resolved CCN measurement, a differential mobility analyzer (DMA) is commonly used to select the particle size before particles enter into the CCN Counter (CCNC). As particles passed through the DMA are not all singly charged under the given electrical mobility, numerous approaches have been proposed to correct multiply charged particles. Multiply charged particles with a larger size penetrate the DMA resulting in a higher CCN activation ratio than the actual value. Higher CCN activation ratio cause error when deriving the critical diameter and, thereby, κ value, a single hygroscopicity parameter (Petters and Kreidenweis 2007). Notably, the multiple charge effect is evident for ambient aerosols that show a large geometric diameter. Frank et al. (2006) proposed a correction method by removing the multiply charged particle fraction in number size distribution scaled by an activation efficiency of an average of five spectra. Petters et al. (2007) used an activation model of CCNC response of transferred polydisperse charged-equilibrated particles that pass through an ideal DMA. The activation efficiency of the particle distribution with the best estimate of critical diameter ($D_c$), diameter to be activated, is determined by minimizing the $\chi^2$ statistic by varying the assumed $D_c$. Rose et al. (2008) simply assumed a constant fraction by of doubly charged particle from the lower level of the plateau in the CCN efficiency spectra. Moore et al. (2010) used the algorithm for scanning mobility CCN analysis (SMCA). The activation fraction of set diameter, $R_a(D_p)$, is determined by removing total particle (i.e., condensation nuclei, CN) and CCN with +2 and +3 charges iteratively until convergence of $R_a(D_p)$. Ultimately, each of methods introduced above are designed to determine $D_c$ of the test aerosols and thus the hygroscopicity parameter, κ, of the aerosols. The single hygroscopicity parameter, κ, is used to model the composition-dependence of the solution water activity. It can be used as a proxy for the chemical composition model and thereby streamline aerosol composition model. Also, the values can manage the hygroscopic properties of complex aerosol types.

Whereas previous studies have improved hygroscopicity retrieval through the development of post-processing algorithms, modern studies have focused on directly manipulating the sampling parameters (e.g. sample flow rate, sheath flow rate, supersaturation, etc.) to allow direct retrieval of κ. Example of this approach include the broad supersaturation scanning (BS2) CCN approach proposed by Su et al. (2016), which modified the inlet and flow system of commercial CCNC to obtain aerosol

hygroscopicity and CCN activity with a high time resolution. With a new monotonic relation between the activation fraction ($F_{act}$) and the activation supersaturation ($S_{aerosol}$), the κ value can be directly determined when the size-resolved CCN activation of set diameter is measured. Kim et al. (2021) implemented a BS2-CCN system and proposed a calibration method. With the CCN activation model, a response of CCNC with and without considering doubly charged particles was compared and suggested a method for the calibration for the BS2-CCN system to minimize the multiple charge effect. Although the

calibration experiment can control particle size distribution to minimize this effect, multiple charging correction method is necessary when it is applied to the ambient aerosol measurement because particle size distribution cannot be controlled and $F_{act}$ directly affects the κ value in the BS2-CCN system. In other words, a high $F_{act}$ value caused by multiply charged particles of ambient aerosols results in an overestimation of κ value. However, several algorithms mentioned above to correct the multiple charged particles cannot be directly applied to the BS2-CCN system. The reasons are as follows: 1) BS2-CCN system

uses multiple and continuous supersaturations in the chamber, not a single supersaturation used in commercial DMT-CCNC. 2) The size-resolved $F_{act}$ value is directly used to derive κ value, whereas original methods focus on a critical diameter or supersaturation. 3) The CCN activation model proposed by Kim et al. (2021) is only applicable to a known calibration aerosol. The model considers the multiple charge effect on total particle and CCN separately, which the equation contains the transfer function for a cylindrical DMA column and the charge distribution of particles carrying elementary charges. Particularly, for

CCN, the function for a fraction of particles that activate as cloud droplets is added into the equation and the information of $S_{aerosol}$ at a specific diameter is necessary. In other words, this model cannot be applied to ambient aerosols with an unknown κ value. Therefore, we propose a multiple charging correction algorithm for the BS2-CCN system in this paper. With a relevant theory for the electric mobility classifier, we summarize the procedure and show examples of ambient aerosol measurement applying the algorithm.

**2 Method**

**2.1 Description of BS2-CCN system**

The BS2-CCN system contains a modified DMT-CCNC with a newly designed inlet system to measure the size-resolved CCN activity with a high-time resolution. The BS2-CCN system includes a differential mobility analyser (DMA), a condensation particle counter (CPC) and a modified DMT-CCNC. Figure S1 and S2 present the schematic plot of a BS2-CCN system and

a newly designed inlet, respectively, which are adopted from Su et al. (2016) and Kim et al. (2021). Selected monodisperse particles by DMA enter into CPC and a modified CCNC, respectively and thereby the size-resolved $F_{act}$ values can be obtained. A new inlet system of a modified CCNC makes a stable low sheath-to-aerosol flow ratios (SAR), which obtain a monotonic $F_{act} - S_{aerosol}$ relation. The aerosol and sheath flow for a modified CCNC are set to 0.46 L min$^{-1}$ and 0.04 L min$^{-1}$, respectively. A detailed description of BS2-CCN system is described in Kim et al. (2021).

## 2.2 Basic theory: multi-charge effect

A particle with a narrow range of electrical mobility ($Z_p$) can pass through the DMA. The $Z_p$ is defined as follows:

$$Z_p = \frac{\varphi e C(D_p)}{3\pi\mu D_p}, \tag{1}$$

Where $e$ is the elementary charge, $\varphi$ is the number of elementary charges on the particle, $C(D_p)$ is the Cunningham slip correction, $\mu$ is the dynamic viscosity of air and $D_p$ is particle diameter. The mobility bandwidth, $\Delta Z_p$, is:

$$\Delta Z_p = \frac{Q_a}{Q_{sh}} Z_p^*, \tag{2}$$

Where $Q_{sh}$ and $Q_a$ are the volumetric sheath flow and aerosol flow, respectively, and $Z_p^*$ is set electrical mobility.

The particle charge distribution at each size can be calculated according to the Wiedensohler (1988) and, Gunn and Woessner (1956). We can calculate the probability that a particle will pass through a DMA classifier using the Kernel function, $G_\varphi(D_p^*, x)$, as follows:

$$G_\varphi(D_p^*, x) = F(x, \varphi)\Omega(x, \varphi, D_p^*), \tag{3}$$

Where $D_p^*$ is a set diameter in the DMA, $x$ is the scale parameter, and $F(x, \varphi)$ is the charge distribution of the particles with $\varphi$ elementary charges, $\Omega(x, \varphi, D_p^*)$ is the probability of particles of $D_p^*$ that pass through the DMA, which is a piecewise linear probability function of triangular shape. It considers the electrical mobility and the mobility bandwidth for a given diameter and flow ratio.

## 2.3 Multiple charging correction algorithm for BS2-CCN

Based on the theory, we present the procedure of a multiple charge correction algorithm for the BS2-CCN system. As the κ value is calculated directly from $F_{act}$, the algorithm aims at deriving true $F_{act}$ we want to obtain from observed $F_{act}$. The experimental setup for the size-resolved CCN measurement is described in Kim et al. (2021). Selected dry particles by DMA split into two ways, CPC for particle number concentration and a modified CCNC for CCN number concentration. It is noted that DMA, CPC and CCNC are controlled and number concentrations of CN and CCN are recorded at once within the own private software. We noted that the algorithm assumes the lognormal size distribution. Scanning the size of particles with an $F_{act}$ of 0 to 1 is performed, and the algorithm applies starting from larger particle to small particle.

1. The CN and CCN number size distribution, $N_{CN}(D_p)$ and $N_{CCN}(D_p)$, are obtained with a time resolution of 1 second. $N_{CN}(D_p)$ and $N_{CCN}(D_p)$ of each particle size were measured for 40 seconds.

2. The size-dependent activation fraction, $F_{act}(D_p) = N_{CCN}(D_p)/N_{CN}(D_p)$, is calculated from CN and CCN time series. For each particle diameter, $F_{act}(D_p)$ is calculated by averaging the values for 25 seconds excluding the first 15 seconds out of the total 40 seconds.

3. To calculate $N_{CCN}$ of multiply charged particles (i.e., Step 5), we need the information of particle diameter of +2 and +3 (or more) charges and corresponding $F_{act}$ values which are not measured. The $F_{act}(D_p)$ values can be obtained

using a linear interpolation within the $F_{act}$ values measured in set particle diameter. The $F_{act}(D_p)$ of particle size larger than the largest $D_p$ is assumed to be 1.0. It is noted that $F_{act}(D_p)$ of largest particle size is to be 1.0.

4.    Starting from the largest aerosol size bin, particle number concentration of single charge that can pass through DMA (set $D_p^*$) is calculated by Kernel function with a given particle number size distribution, $n(x)$ after removing the number concentration of particles with +2 and +3 (or more) charges using Eq. (4). $D_{max}$ should be larger than the

diameter that can cover the Kernel functions with different charges (Here, $\varphi = 1, 2, 3$) under the set particle diameter range. For example, peak diameter with a +3 charge in the Kernel function is about 131 nm when the particle size is fixed at 70 nm in DMA (Fig. 1).

$$N_\varphi(D_p^*) = \int_0^{Dmax} G_\varphi(D_p^*, x) n(x) dx, \tag{4}$$

5.    As CCN number concentration can be calculated with the product of $F_{act}$ and $N_\varphi$, we can calculate true $F_{act}$ value,

$F_{act}^*$, by subtracting the CCN number concentration by multiply charged particles from the measured CCN number concentration. To be specific, CCN number concentration of multiply charged particles can be calculated with information of particle diameter of +2 and +3 (or more) charges and corresponding $F_{act}$ from $F_{act}$ curve (Eq. 5). It is noted that $h(x, \varphi, D_p^*)$ is a mathematical form of an activated fraction, $F_{act}$, function for $D_p^*$ with $\varphi$ elementary charges.

$$N_{CCN}(D_p^*) = \sum_{\varphi=1}^{3} \int_0^{Dmax} G_\varphi(D_p^*, x) h(x, \varphi, D_p^*) n(x) \, dx \, d\varphi, \tag{5}$$

6.    The calculated $F_{act}^*(D_p^*)$ that we want to obtain in step (5) is updated to the existing $F_{act}$ function, $h(x, \varphi, D_p^*)$.

7.    Repeat step (3)- (6) from the largest to the smallest diameter.

Through the single process, Step (3) – (6), we obtain the $F_{act}^*(D_p^*)$ value, and the calculated $F_{act}^*(D_p^*)$ is reflected into the $D_p -$ $F_{act}$ curve which is used in the calculation process of the following particle size.

Figure 1 shows the calculated Kernel function for the DMA (TSI 3080) set the size of 70 nm with different charges ($\varphi = 1, 2, 3$). It is noted that particles carrying more than three charges within the particle size range in ambient air are not considered in this study. The sample/sheath flow ratio of 1.5/10 is applied for calculation, the same setting for calibration experiment and ambient aerosol measurement, mainly discussed in Section 4. Kernel function can be different with the sample/sheath flow ratio. Particle number concentration carrying a specific charge can be determined by Kernel function and particle number size

distribution (Eq. 4). Figure 2 shows particle number fraction at each charge under an assumed particle distribution data obtained from ambient aerosol measurement. According to Fig.2, most particles are composed of particles with a single or double charge, but the particle number fraction by triple charge between 60 nm and 90 nm, where the particle number concentration is high, was also slightly seen. The number concentrations by singly charged particles were dominant, but there was a size range where a number fraction by doubly charged particles was more than 0.1. And, the number fraction of doubly and triply charged

particles depends on the number concentration of the larger particles. It means that in a region where larger particles are

dominant, the effect of a doubly charged particle would be more prominent, and thereby it is necessary to apply the multiple charging correction algorithm.

## 3 Application

The multiple charge correction algorithm is applied to the calibration experiment and ambient aerosol measurements. Before applying the algorithm, a log-normal fitting procedure for aerosol number concentration is conducted to calculate the number concentration of particles by double and triple charges as described in Eq. (4) and (5). It is well known that the size distribution of atmospheric aerosols can be expressed by a log-normal distribution, and other fitting functions can be used if the number size distribution cannot be described by the log-normal distribution.

### 3.2 Calibration experiment

Calibration experiments for the BS2-CCN system are performed with ammonium sulfate, well-known calibration aerosols for CCNC under the dT= 6 K and 8 K conditions. An instrumental setting of the calibration experiment is the same as the existing BS2-CCN system except that an atomizer for generating a calibration aerosol is added to the front of the DMA. Details are described in Kim et al. (2021). Figure 3 shows the aerosol and CCN number size distribution, activation fraction ($F_{act}$) curve

and BS2-CCN calibration curve under the two dT conditions. According to Fig.3a, CCN number size distribution shows a small plateau in the front by multiply charged particles leading to the small plateau of $F_{act}$ (Fig. 3b). When applying the algorithm, this small plateau disappears, and the other $F_{act}$ value slightly decrease. The results show that the algorithm corrects the increased $F_{act}$ values due to multiple charge particles. The elevated $F_{act}$ values induced by multiply charged particles in Fig.3 are not that high, less than 0.1, due to the small geometric mean diameter ($D_g$) and standard deviation ($\sigma_g$) of a generated

particle number size distribution. It is the way to minimize the effect of multiply charged particles on the calibration curve (Kim et al. 2021). Figure 3a and Table S1 show the calibration curve, coefficients of the fitted calibration curve, and goodness of fit before and after the correction. There is no significant difference in each coefficient and the shape of the calibration curve. The reasons for this are as follows: 1) For the fitting procedure, values lower than 0.05 of $F_{act}$ value where small plateau by multiply charged particle exists are excluded. 2) Calibration experiment presented in this section follows the

recommendation suggested by Kim et al. (2021) to generate particle size distribution with small $D_g$ and $\sigma_g$ and thereby, the effect of multiply charged particles is not that obvious. 3) $F_{act} - S_{aerosol}$ relation is determined based on $\kappa - K\ddot{o}hler$ theory (Petters and Kridenweis, 2007). In other words, the value of $S_{aerosol}$ changes based on the theory according to the reduced $F_{act}$. It can be inferred from these results that the suggested algorithm corrects the $F_{act}$ induced by doubly and triply charged particle well. In addition, a more accurate calibration curve can be obtained by applying the algorithm, but the effect of multiply

charged particles can be minimized, and almost the same calibration curve can be obtained without correction if the experimental control is performed well. Kim et al. (2021) examined the effect of multiply charged aerosols on calibration

curve through CCN activation model and experiments. By adjusting the particle number concentrations, number size distribution with a small $D_g$ and $\sigma_g$ could be generated to minimize the influence of multiply charged particles. Additionally, Fig. S3 and S4 show the combined result of ammonium sulfate calibration experiments for three dT conditions and sodium chloride experiment, another representative calibration aerosol. All show consistent results with the Fig.3 indicating that the suggested algorithm performs well.

### 3.3 Ambient aerosol

Unlike calibration aerosols, particle number size distribution of ambient aerosols cannot be controlled and $D_g$ is generally larger than lab-generating aerosols. Also, particle number size distribution is highly variable depending on time and region. It means that $F_{act}$ can be easily affected by multiply charged particles, so the correction algorithm is essential. Here, we adopted cases from the polluted marine aerosol measurement that was held in the Yellow Sea during springtime, 2021. Specifically, the ship campaign was held three times, about 10 days each (1st period: 22 March – 2 April, 2nd period: 6 – 15 April and 3rd period: 20 – 29 April), and crossed the north and south of the Yellow Sea alternately. During the first observation period, ship went down to the South (31.5 °N, 125.0 °E), which was relatively less affected by the continent. Also, there was a period for the transportation of Asian dust. Among the first observation period, we chose two cases with different aerosol size distributions; one from the period when the ship sailed south and one from the period when Asian dust was transported. As in Section 4.1, the measured aerosol number concentration data were fitted to the log-normal distribution for calculation and the aerosols are assumed to be spherical. Figure 4 shows the measured aerosol and CCN number size distribution and the activation curve before and after the correction of Case I. $D_g$ and $\sigma_g$ are 70.6 nm and 1.58, respectively. When applying the algorithm, the reduction in the $F_{act}$ value of the activation curve is 0.01 to 0.07. In particular, the decrease was the largest at about 70 nm, near the geometric mean diameter of the aerosol size distribution. Figure 5 is the same as Fig. 4 with Case II, larger $D_g$ and higher $\sigma_g$ than Case I ($D_g$ = 129.0 nm and $\sigma_g$= 1.65). As the peak diameter becomes larger, the effect of multiply charged particles increases. The $F_{act}$ decreases by up to 0.2. Although $N_{CCN}$ by doubly charged particle shows the maximum near the peak of $N_{CN}$, the decrease in the value of $F_{act}$ was dominant in the range where $F_{act}$ is 0.1 to 0.8, and the corresponding particle size was smaller than the peak diameter. It is because $F_{act}$ is calculated not only by $N_{CCN}$ but also by $N_{CN}$ after subtracting multiply charged particles. Also, when calculating $N_{CCN}$, the original $F_{act}$ from the measurement is considered. Figure 6 shows the change of κ value according to the change of $F_{act}$. In Case II, since the decrease in $F_{act}$ is more prominent than in Case I, the difference in κ value after the correction is also more significant. For Case II, the change in the κ value is about 0.04 to 0.08, whereas for Case I, the change in κ is about 0.01. We can conclude from these results that the effect of multiply charged particle is different with particle number size distribution. And, the difference in κ of 0.08 can lead to a relative deviation of CCN number concentration of up to about 20% when assuming the particle size distribution of Case II and supersaturation ranging from 0.05 to 1.0%. It means that the correction algorithm should be applied if the peak size of ambient aerosols is large and has a broad particle number size distribution. However, it is noted that not only monomodal distribution but also

bimodal/trimodal aerosol distribution are observed in the ambient environments, and care should be taken to apply the
correction algorithm accordingly.

**4 Discussion and conclusion**

As discussed above, $F_{act}$ by multiply charged particle can affect the activation spectra and thereby the κ derivation. The extents of $F_{act}$ and κ deviation between original and corrected activation spectra (i.e., $D_p - F_{act}$) for Case I and Case II were different due to the different particle number size distribution. Therefore, we further examined how particle number size distribution is linked to the deviation of activation spectra and κ values. Based on the observed activation spectra, referred to as 'original', we applied the correction algorithm by changing $D_g$ of particle size distribution. The value obtained by applying the algorithm refers to as 'corrected'. The relative deviation is defined as the difference between the original and corrected value. Figure 7 is a contour plot of the relative deviation of $F_{act}$ and κ as function of $D_g$ for the log-normally fitted particle size distribution for Case I and Case II. The red dashed line indicates the original $D_g$ of each Case as described in Section 3.3. For Case I (Fig.7a), the maximum value appears near the original $D_g$, 70nm, and gradually decreases. As $D_g$ increases, the relative deviation of $F_{act}$ increases, and the interval between the contour also gradually narrows. It is noted that deviation values between 60 and 120 nm in $D_p$ which correspond to $F_{act}$ value between 0.1 and 0.9 are meaningful. The deviation of κ value in Fig. 7b also showed a similar tendency to that of $F_{act}$ value. For Case II (Fig. 7c), the maximum relative deviation value appears near the 60 nm. The relative deviation value of $F_{act}$ is higher and the interval is narrower than Case I. It can be explained by the original particle number size distribution and activation spectra in Fig.4 and 5. Case II shows a much broader and larger particle size distribution, which means the effect of multiply charged particle is larger in Case II. Also, $F_{act}$ shows a higher value at the same particle size in Case II than Case I implying different chemical composition due to different air masses. For Case II, deviation values between 50 and 100 nm in $D_p$ which correspond to $F_{act}$ value between 0.1 and 0.9 are meaningful. Like Case I, Case II also shows that the deviation of $F_{act}$ increases as $D_g$ increases, and the deviation of κ also tends to be similar (Fig.7c and 7d). It means that change in particle size distribution affect the deviation of $F_{act}$ and thereby κ value regardless of the original activation spectra.

In addition to the effect of multiply charged particles on CCN activation, there are other possible uncertainties of CCN activation in the BS2-CCN system. It can be affected by combined physical factors including mixing state, surface activity, viscosity and particle morphology. For example, CCN number concentration can be underestimated in the ultrafine mode with a high organic mass fraction due to the lower water surface tension. Particle morphology can affect CCN activity as it can influence in determining the diameter corresponding to the centroid mobility of each size bin. Altaf et al. (2018) explored the effect of size-dependent morphology on CCN activity using aerosol particles composed of organic compounds and salts. For calibration experiment, particle morphology is considered by applying the particle shape factor when deriving κ values. The relative uncertainties, the ratio between absolute error and measured value, do not exceed 1% for calibration aerosols. However, the relative uncertainties could be higher in ambient aerosols due to the non-sphericity. As κ values can be derived directly by

$F_{act}$ values in the BS2-CCN system, these possible uncertainties of CCN activation need to be carefully considered when deriving κ values. For BS2-CCN system, the absolute deviation of $F_{act}$ was mostly less than 0.05 except for the stabilization time, the first 10 seconds for each $D_p$ scan and the $F_{act}$ value that is higher than 0.85 (Kim et al. 2021).

In summary, we propose a multiple charging correction algorithm for the BS2-CCN system with continuous and multiple supersaturations in the chamber. Unlike existing algorithms, the correction algorithm in this study aims at deriving the true value of $F_{act}$ at each particle size as $F_{act}$ value is used to derive κ directly. Starting from larger particles, number concentrations of particles both $N_{CN}$ and $N_{CCN}$ with multiple charges (here, we consider only +2 and +3 charges) are removed, and new $F_{act}$ is updated to the original activation spectra. The same procedure is repeated to the smaller particle after updating new $F_{act}$ value. When applying the algorithm, the $F_{act}$ value was corrected, especially in the range between 0.1 and 0.9 of $F_{act}$ and the plateau in activation spectra by multiply charged particles was successfully eliminated. The extent of correction depends on the particle size distribution. To be specific, there is no large discrepancy of activation spectra and calibration curve (i.e., $F_{act} - S_{aerosol}$ relation) if the particle size distribution of generated calibration aerosols is well controlled with smaller $D_g$. For ambient aerosols, larger deviations of $F_{act}$ and κ were shown with larger $D_g$. The difference between corrected and original κ values of 0.08 from samples could lead to about 20% of the relative deviation of CCN number concentration within the supersaturation range between 0.05 and 1.0%, which cannot be ignored. It can be concluded that the multiple charge correction algorithm should be applied to the ambient aerosols to reduce the error when calculating the κ value in the BS2-CCN system.

**Code and data availability**

Algorithm code and data can be available upon request from the corresponding author (yafang.cheng@mpic.de).

270 **Author contributions**

NK designed and implemented the data correction algorithm for BS2-CCN system. YC supervised and led the paper preparation. HS and NM contributed to the discussion with their expertise in BS2-CCN system and measurement data processing. NK wrote the paper. All coauthors discussed and results and commented on the paper.

**Competing interests**

275 At least one of the (co-)authors is a member of the editorial board of *Atmospheric Measurement Techniques*.

**Acknowledgments**

This work was supported by the Max Planck Society (MPG). Y.C. and N.K. thank the support from Minerva Program from MPG.

280

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

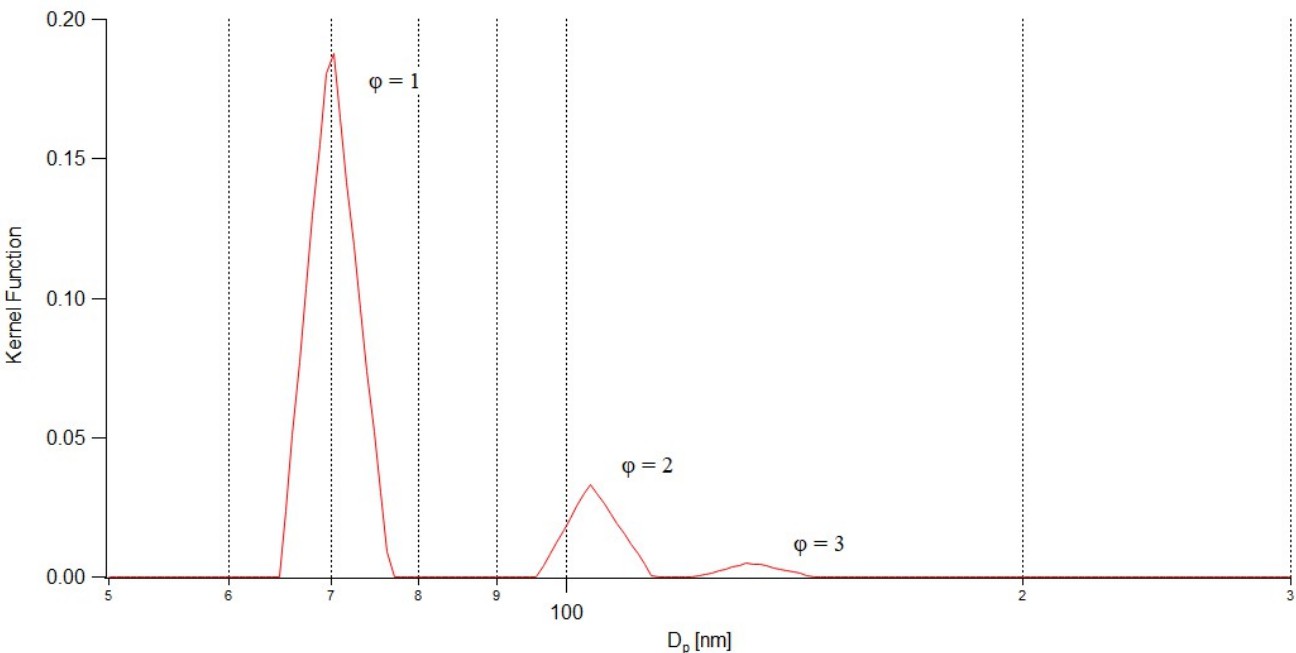


**Figure 1: Kernel function for the DMA set size of 70 nm with different charges ($\varphi = 1, 2, 3$). The sample/sheath flow rate is 1.5/10.**

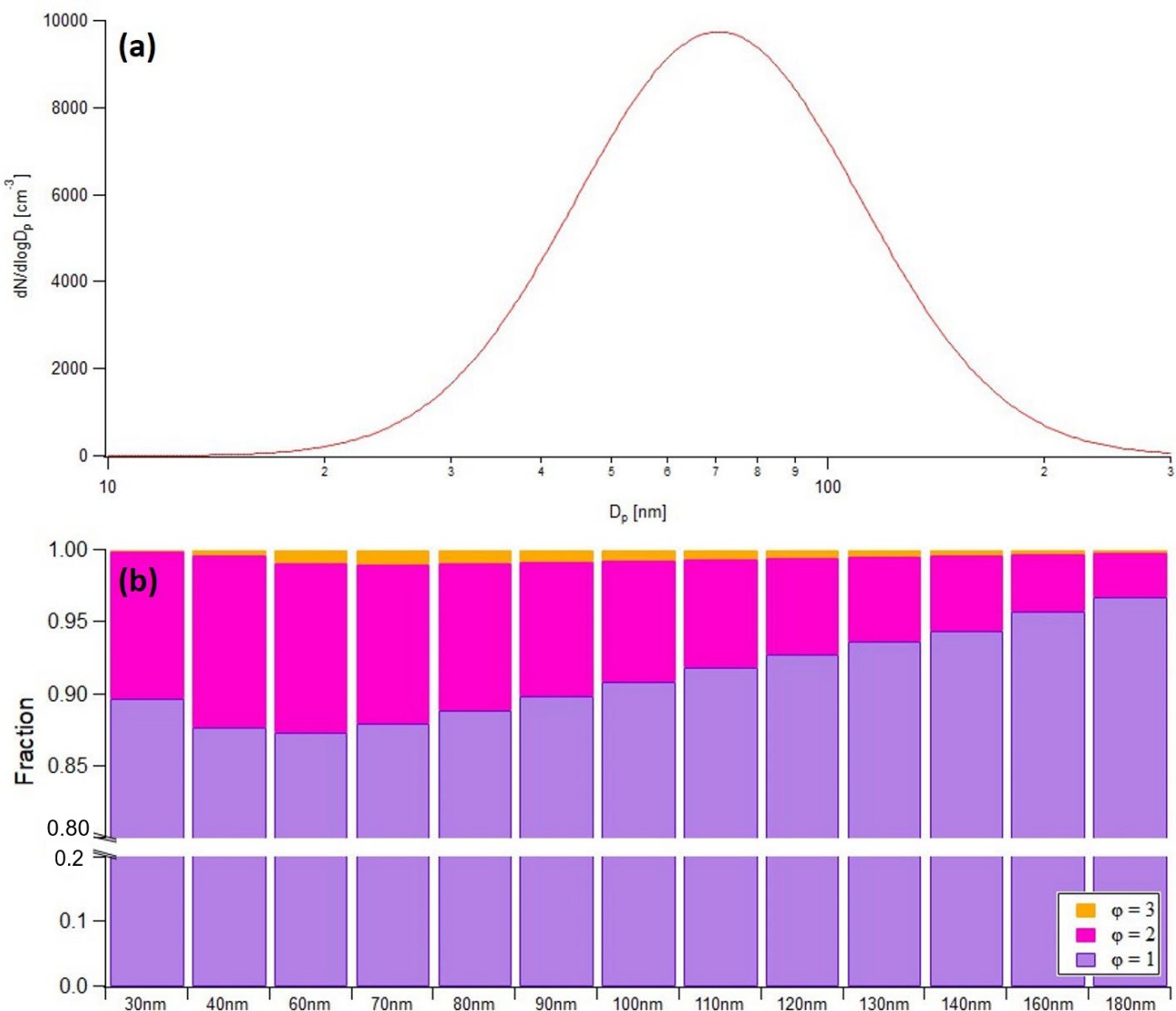

Figure 2: (a) Lognormally fitted particle number size distribution ($D_g = 70.6\,nm, \sigma_g = 1.57$) and (b) particle number fraction at each charge to which the particle number size distribution is applied. Each color indicates charge number: Purple for singly charged particles ($\varphi = 1$), pink for doubly charged particles ($\varphi = 2$) and orange for triply charged particles ($\varphi = 3$).

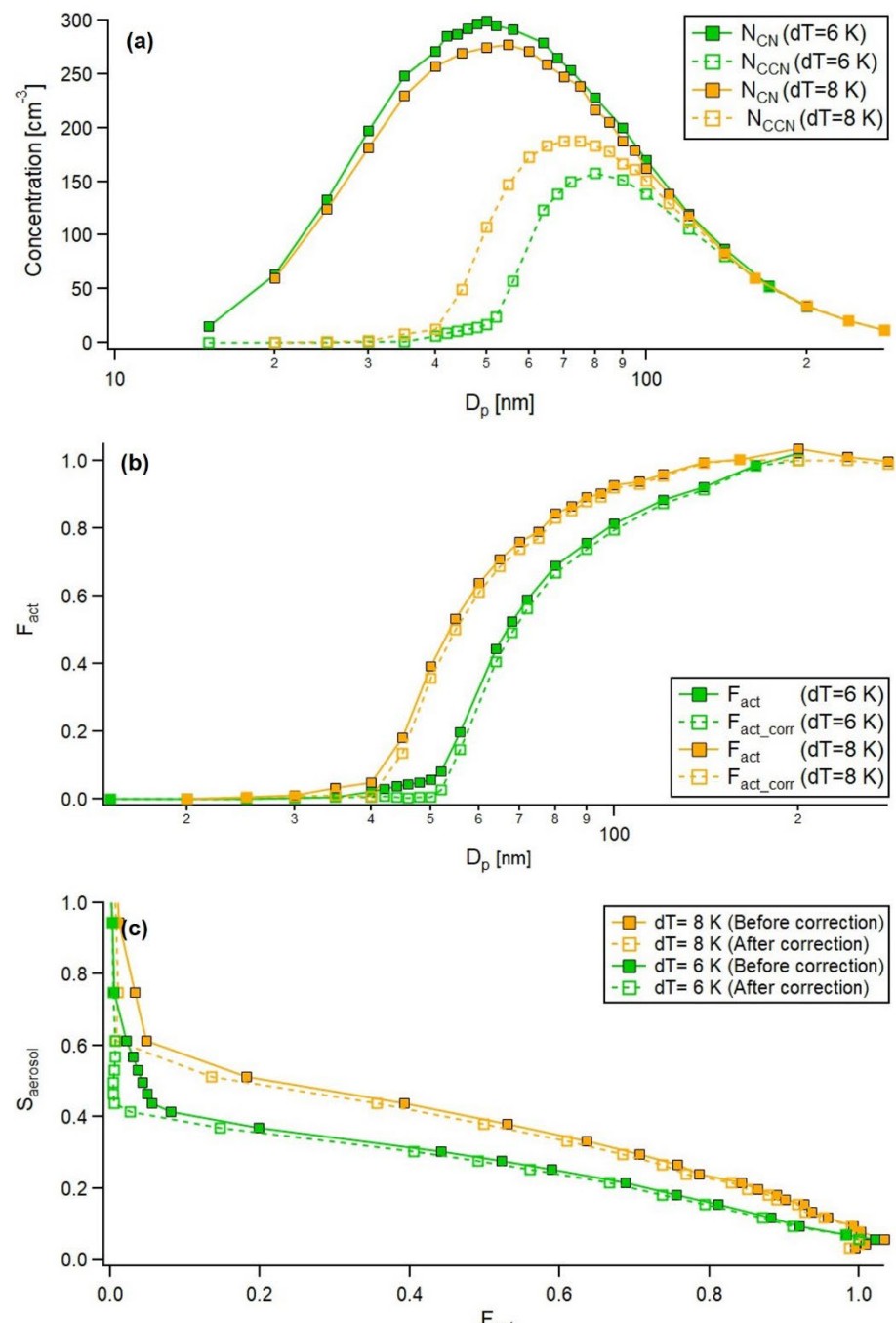

**Figure 3: (a) Particle and CCN number size distribution (b) activation fraction ($F_{act}$) curve and (c) $F_{act} - S_{aerosol}$ calibration curve for dT = 6 K (green) and 8 K (orange). Squares indicate measurement points. Solid line and dashed line with squares indicate results before and after the correction, respectively.**

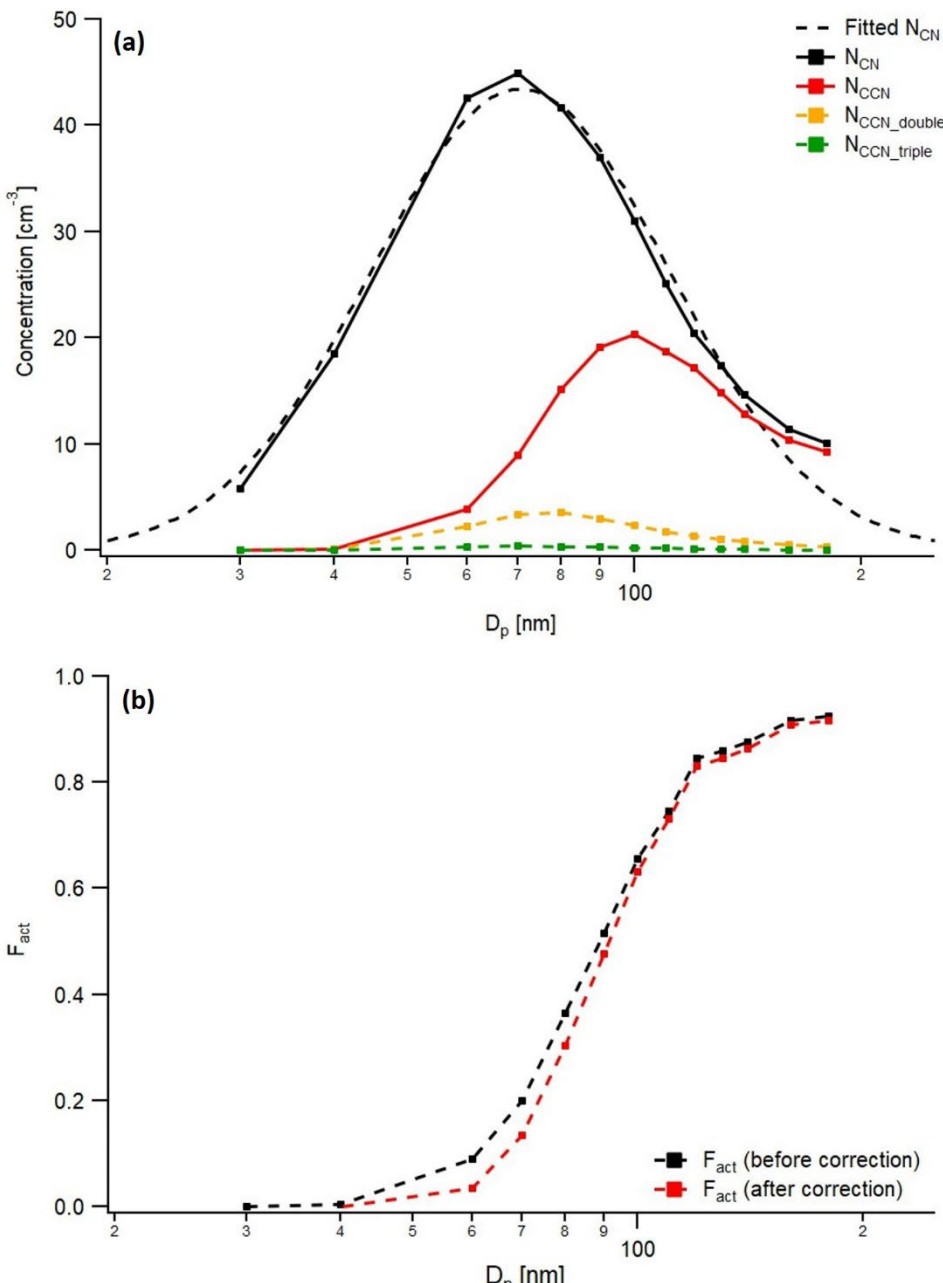

**Figure 4: (a) Number size distribution of aerosol (black) and CCN (Red for total particle, Orange for doubly charged particle and Green for triply charged particle) and log-normally fitted particle number size distribution ($D_g$ = 70.6 nm and $\sigma_g$ = 1.58, black dashed line), and (b) activation cure before (black) and after correction (red) for Case I. Each dot indicates measurement point.**


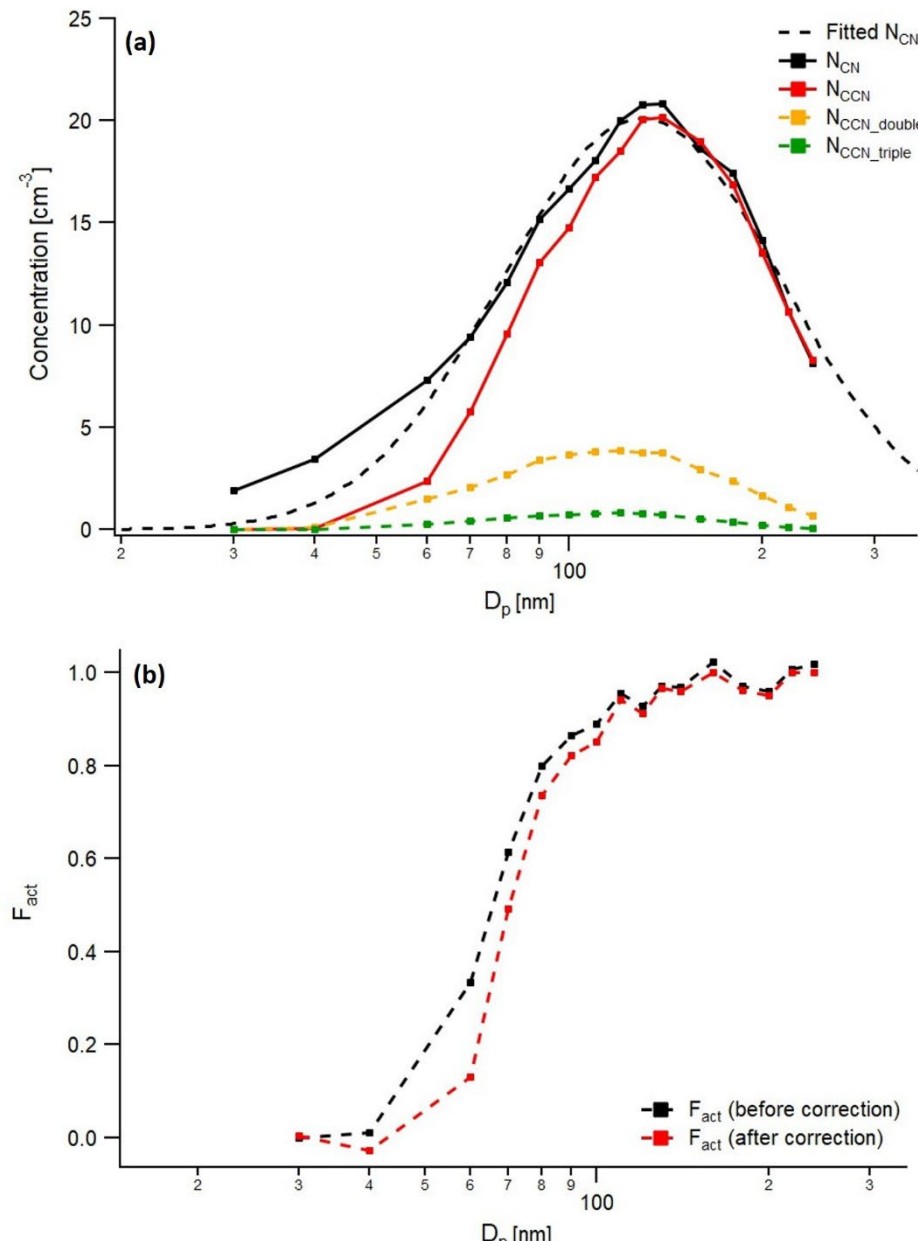

**Figure 5: (a) Number size distribution of aerosol (black) and CCN (Red for total particle, Orange for doubly charged particle and Green for triply charged particle) and log-normally fitted particle number size distribution ($D_g$ = 129.0 nm and $\sigma_g$ = 1.65, black dashed line), and (b) activation cure before (black) and after correction (red) for Case II. Each dot indicates measurement point.**

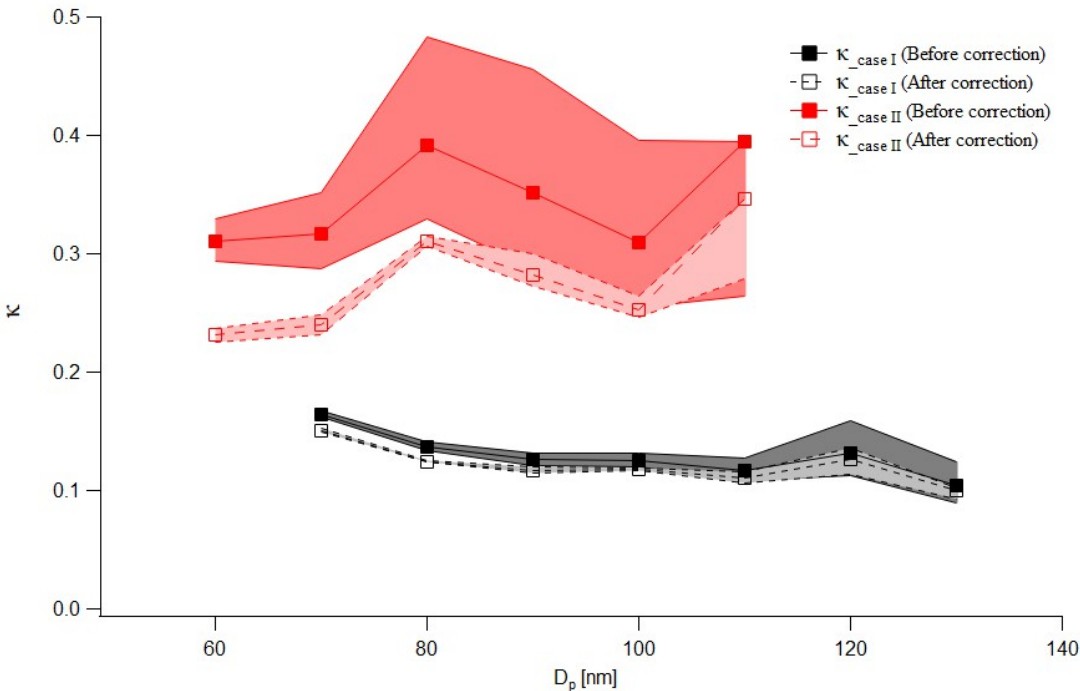

**Figure 6: κ values as function of particle size before (filled square) and after (square) correction for Case I (black) and Case II (red). Shaded area indicates uncertainties of κ values which is evaluated from the $F_{act}$ values.**


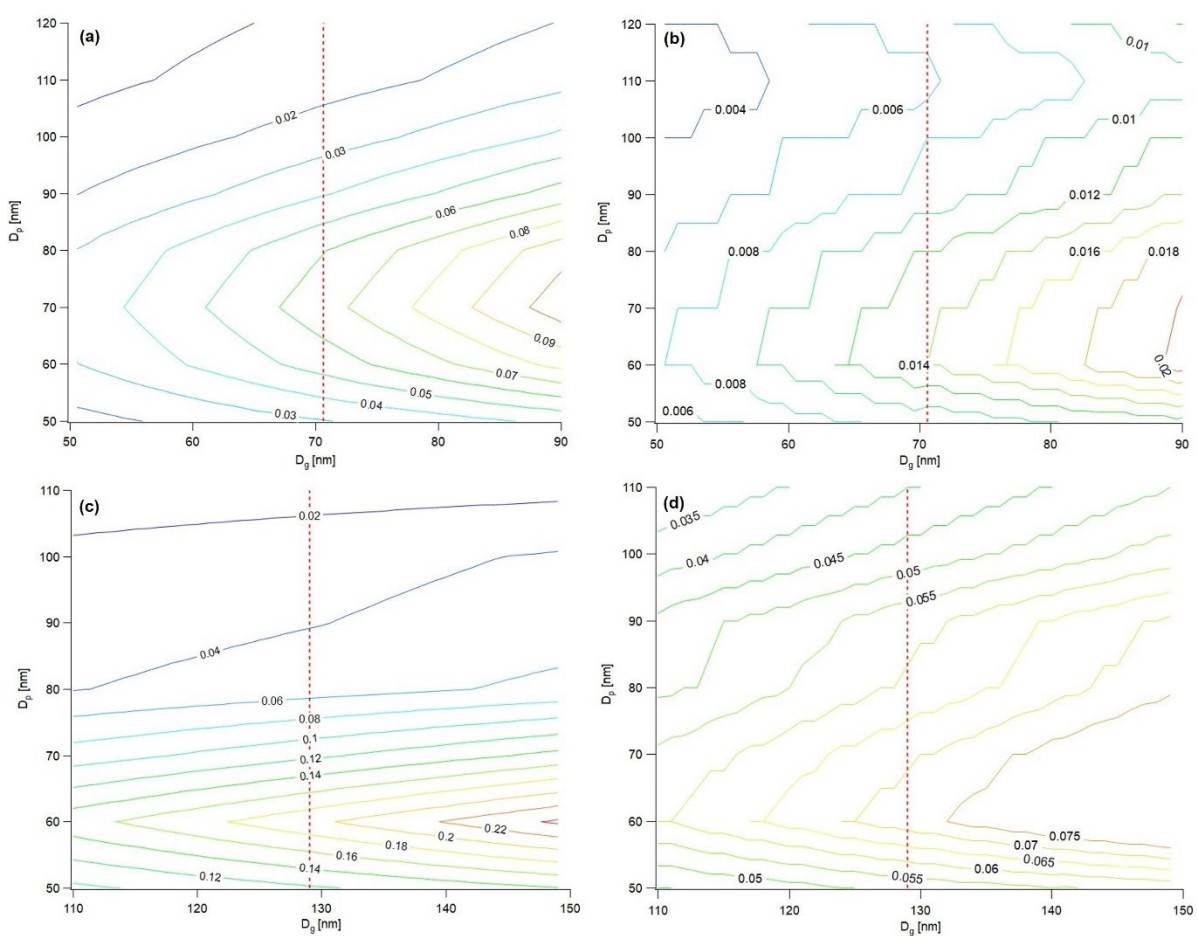

**Figure 7: The deviation (Original value – Corrected value) of $F_{act}$ ((a) for Case I and (c) for Case II) and κ ((b) for Case I and (d) for Case II) as function of $D_g$ for log-normally fitted particle size distribution. Contour indicates the range of relative deviation and the value is marked on the contour (The redder the color of the contour, the greater the value). Red dashed line presents the original $D_g$ value for each case ($D_g$= 70.6 nm for Case I and $D_g$ = 129.0 nm for Case II).**