# Peer review of "A multiple charging correction algorithm for broad supersaturation scanning cloud condensation nuclei (BS2-CCN) system"

_Atmospheric Measurement Techniques, 2022_

## Referee Comment (RC2)

**General Comments**

This article presents a new algorithm for processing measurements from a Broad Supersaturation Scanning Cloud Condensation Nuclei (BS2) system. The outcome of this algorithm is to improve retrieval of aerosol hygroscopicity parameters, namely the particle hygroscopicity, $\kappa$. The article claims that this algorithm provides a unique solution to a known problem in determining $\kappa$: namely, that multiply-charged particles that pass through the differential mobility analyzer (DMA) in a BS2 system result in misshapen particle activation curves which degrade the retrieval of $\kappa$. Despite the claim of novelty, the algorithm bears a rather close resemblance to the proposed methodology of Moore et al (2010).

In general, the Methods section is missing sufficient detail for their method to be utilized and reproduced by other researchers. Some issues are purely technical: the authors need to re-work the notation of the Methods section. There are several instances where the notation is not appropriate, misleading, or definitions are missing altogether. Other issues are pragmatic: further descriptions of their BS2 system should be included (rather than referenced) such as impactor size, DMA size detection range, etc. This is, essentially, a methods paper and the Methods section is perhaps the weakest point of the current manuscript. It should be spelled out to the letter what a researcher needs to do to implement this method.

The results demonstrate a fulfillment of the original promise. The appearance of multiply charged particles in the activation curve have disappeared. However, the assessment of this methodology is fairly qualitative. The case study approach is not sufficient enough argument for researchers to understand when this correction needs to be applied to their measurements. It is clear, for example, that the correction algorithm need not be applied to calibration experiments. A revision of this manuscript should include a more quantitative laboratory-based study with ammonium sulfate rather than the qualitative field-based study that is currently used. The revision should also include a full uncertainty analysis to determine confidence intervals on derived hygroscopicity. This would allow other researchers to better understand when they should apply this correction and the magnitude of the effect on hygroscopicity retrieval (so that they can troubleshoot their implementation). The authors should also make it more apparent in the abstract and conclusions that the proposed algorithm assumes that the particle size distribution is monomodal. The algorithm has not been tested for more complicated PSDs.

Finally, the authors should support their claim that their methodology is a necessary improvement on previous approaches. A revision of this manuscript should also include a side-by-side comparison of this methodology to previously proposed methods in the literature, e.g. Moore, Nenes & Medina (2010) to which the proposed method bears an uncanny resemblance.

**Specific Comments**

**Introduction.** The introduction does a good job of introducing source material and identifying where this manuscript is positioned within the scientific literature. However, the importance of obtaining the hygroscopicity parameter, k, should be introduced much earlier so as to better guide the narrative flow of the rest of the introduction. Currently, the introduction seems very technical with no obvious goal.

**Methods.** There are many minor and major technical issues in the Methods section that must be fixed. In general, \mMore detail is needed such that others can apply this method to their work. By section & subsection:

**Section 2.1** The use of "$\upsilon$" as the number of elementary charges is unusual and confusing. The variable "$\upsilon$" ought to be reserved for kinematic viscosity in this context. Some authors use $\varphi$ to represent the number of charges, e.g. Collins, Flagan & Seinfeld (2003). Next, the authors omit the definition of the set mobility, $Z^{*}_{p}$, which is not at all equivalent to $Z_p$. This might not be obvious to a reader at first pass, so $Z^{*}p$ must be defined. Third, it may be more correct to say that Wiedensohler (1986) developed an empirical model based on charging theory of Fuchs (1963). Additionally, Wiedensohler's model is only valid up to ±2 charges. Beyond this number of charges, it is common to use Gunn (1956). As you evaluated charging probabilities for +3 charges, you would need to apply the formulae described by Gunn.

To avoid confusion $G_v$ should be redefined:

$$G_v\big(D_p, x\big) = F(x, v)\Omega(x, v, D_p)$$

Without the summand. It should also be specified how $\Omega(x, v, D_p)$ was calculated. What processes/efficiencies are involved in its calculation? Penetration efficiency? Impactor Efficiency? Diffusive losses? This is a methods paper. More detail is needed such that others can use this tool.

**Section 2.2**. I am wondering if it needs to be specified if the data are collected simultaneously on a single computer or if two separate computers are used and the data are analyzed offline. If the latter is the case, there must have been some need to make sure that the clocks were aligned for 1 Hz measurement. If so, specify!

**Step 1.** Are these variables named correctly? The use of C implies that these are counts, not concentrations (N). Further, you should be more specific about what algorithm is used to compute the inversion. It is not sufficient to say that private software is used (do you mean the AIM software?). In Section 3, you also mention that a lognormal size distribution is assumed in this step. If it is, specify this here.

**Step 2.** As written, this is open to misinterpretation. Is the DMA set at a "single" particle size for 40 s, you omit the first 15 s and average the latter 25 s? Or, are you saying that the DMA is set at a "single" particle size for 25 s, you omit the first 15 s, and average the last 10 s? Clarify please.

**Step 3.** There is more than one way to interpolate between two points... linear? Polynomial? Spline? Be specific! It is also worth asking: what is the goal of interpolation at this step? Is it to determine h(x, v, D$_p$)? If so, specify. Additionally, it's not obvious that the activated fraction should depend on the number of charges. In my later comments, I rearrange your integral.

**Step 4.** Did you actually integrate Equation (4) out to infinity? I am guessing not... It is common to integrate up to some factor of the impactor cut-off, D$_{50}$. Specify the actual upper bounds of your integration. Additionally, I believe that Gv is incorrectly defined. Based on what this step says, I assume that there was an intermediary step of calculating the following:

$$G_{+2}(D_p, x) = F(x, +2)\Omega(x, +2, D_p)$$

This is different than the current definition of G$_v$, which would include all charges as defined in the manuscript (See my comments on Section 2.1). After calculating G$_{+2}$, I assume that one would multiply G$_{+2}$ by the proposed solution spectrum n(x) and integrated to retrieve the number of doubly-charged particles of size Dp, N$_{+2}$(D$_p$), i.e.:

$$N_{+2}(D_p) = \int_0^\infty n(x)G_{+2}(D_p, x)dx$$

Either re-define G$_v$ or be more explicit.

**Step 5:** Again... What is the upper limit of charges that were considered? You state that no more than 3 charges are calculated, but it should be specified in this step. I think this is a justifiable cut-off step given the limited range of 300 nm, but you should be more quantitative in your argument for why you terminated the sum at +3 (e.g. less than X% of particles have charge >3 at the impactor cur-off diameter). Additionally, charge is discrete, so integral notation is inappropriate. This should read:

$$N_{CCN}(D_p) = \int_0^\infty n(x)h(x, D_p)\sum_{v=1}^{+3} G_v(D_p, x)dx$$

At this stage in my review, I am beginning to wonder how different this procedure is from the SMCA proposed by Collins et al (2010). Is it unique because you are using slightly different flow rates and supersaturation settings? Or is it unique according to Step 5?

**Application**

As noted above, the log-normal fitting procedure should be described in the Methods, not the results.

**Section 3.2.** What is dT? Spell it out, please. You should also be more descriptive/quantitative of your test distribution. What was the value of the geometric mean diameter and **geometric** standard deviation? Even if the reader can go read your other paper (Kim, 2021), re-iterate the suggested limits of $D_g$ and $\sigma_g$ for a "well-performed" calibration experiment here. Finally, Table 1 could probably be added to a Supplemental Information document, I don't think there is any benefit to it appearing in the main text.

**Section 3.3.** It should be noted that that ambient aerosol size distributions are rarely monomodal, like this retrieval assumes. The authors should thus comment on what environments, if any, it is safe to assume that the distribution is monomodal. In the marine environment, the particle size distribution is often at least tri-modal.

The referral of the reader to another paper for information about your campaign is frustrating. Basic information like the duration of the campaign should be specified. The selection of cases should also be detailed and better motivated. It is clear after reading this section that you want to highlight the degree to which $D_g$ determines the magnitude of the effect your multiple-charge correction will have, but this should be spelled out at the beginning of the section.

I also believe that this effect could have been better studied directly in the lab with particles of known hygroscopicity, e.g. ammonium sulfate, by modifying the atomizer pressure & solution concentration. This would allow you to directly relate the deviation in k as a function of Dg. Uncertainty analysis could then allow you to determine a threshold Dg below and above which your correction should/shouldn't be applied. This would provide your readers with a much more satisfying quantitative assessment than the relatively qualitative case study that is presented.

Next, $F_{act}$ is a function of particle size. Are you saying that the average value of $F_{act}$ decreased from 0.01 to 0.07? Or, are you referring to a specific value of $F_{act}$ at a diameter of 60 nm? Follow-up question: How can something decrease from 0.01 to 0.07?

I would like to see a proper error analysis added to this section. It is clear that your algorithm results in changes to retrieved values of hygroscopicity, k. It also seems to complete the task as advertised, removing multiply charged particles from the minor plateau, but are these changes significant? Add 95% C.I. to your estimates of k in Figure 6. This will help quantitatively reinforce your claim that multiple-charge corrections need only be considered when the peak of the observed size distribution is >100 nm.

Finally, this manuscript is staking a claim that the algorithm uniquely solves the multiple-charge correction in the context of SMPS+CCN measurements. It would be useful to actually demonstrate that this is true, since the algorithm bears such close resemblance to the Collins, Nenes & Medina (2010). The algorithm seems to predict the greatest change when Dg is >100 nm. It would be

useful to do a side-by-side comparison of data processed by the proposed algorithm and existing algorithms in the literature.

**Discussion.** Uncertainty/Error analysis should also inform the discussion of Figure 7. This would help support your claim that these deviations are meaningful by demonstrating that they are statistically significant.

**Final Note:** As it currently stands, you have demonstrated that your correction algorithm works for monomodal aerosol. It is a rather large stretch that you finish this manuscript by saying it should be applied to "a variety of particle number size distributions." Remove this statement or prove it.

**Technical Corrections**

Line 28:        " **a** key element". I disagree with the notion that clouds are the sole element controlling climate change.

Line 29:        Change to "Despite the scientific importance of CCN,"

Line 30:        "aerosol-cloud interaction**s**". There are many types of aerosol-cloud interactions.

Line 31:        Citation needed at the end of "composition and processes." Perhaps the updated version of this figure:
https://archive.ipcc.ch/publications_and_data/ar4/wg1/en/tssts-2-5.html.

Line 33:        "" → "in recent"

Line 40:        "under the simple assumption" Are you referring to a specific assumption here? ZSR? Or, did you mean to say "under **this** simple assumption."

Line 51:        On first introduction, explain what $D_c$ is.

Line 52:        "Constant fraction  **of** doubly charged particle**s"**

Lines 54:        Not sure if it is worth mentioning that the process starts from the largest aerosol size bin and iterates towards smaller bins.

Line 55:        Suggested rephrasing: "Ultimately, each of the methods introduced above are designed to determine the critical activation diameter, Dc, of the test aerosol and thus the hygroscopicity, k, of the aerosol." Additionally, it should be outlined at the beginning of this paragraph that the hygroscopicity parameter, k, is the desired outcome.

Line 57:        You should introduce the theme of this paragraph in the first sentence. "Whereas previous studies have improved hygroscopicity retrieval through the

development of post-processing algorithms, modern studies have focuses on directly manipulating the sampling parameters (e.g. sample flow rate, sheath flow rate, supersaturation, etc.) to allow direct retrieval of k. Examples of this approach include…"

Line 58:     What is the calibration experiment? Either describe or omit.

Line 68:     Do you mean continuous as opposed to discrete? Or continuous as in "temporally continuous". It might be beneficial to describe as "continuously variable".

Line 82:     What is Dp? Be thorough and describe.

Line 84:     This equation is valid, but what is Z*p?
$$Z^*_p = (Q_{sh}/(2*\pi*V*L))*\ln(r_2/r_1)$$

---

## Author Comment (AC1)

**Response to Reviewer #2**

Overall, the work is new and the method to apply multiple-charging corrections for the BS2-CCN instrumentation has not been done before. The work extends on previously published works (e.g., the kernel function is added to the Kim et al 2021 multiple chare correction) with the BS2-CCN data. In general, the main concern is that multiple factors affect CCN measurement (aerosol shape, aerosol aggregation, viscosity, volatility, solubility, surface activity) and these effects are confounded; it is difficult to isolate the effects of multiple charging alone. However, the data collected from the BS2-CCN counter and subsequent analysis will be important for understanding CCN spectra of atmospheric aerosol and thus the work warrants publication. The following questions and comments address ideas that maybe unclear to the reader in the manuscript.

A: We thank the reviewer for encouraging and helpful comments on our manuscript. We believe that the quality of our manuscript is improved as we reflect the reviewer's comments. Below each of the questions/comments is written with the Italic font and then our response is followed with the normal font.

Comments

*Q1: Composition of aerosols does not affect the probability of multiple charging. However, the morphology of the aerosols changes the probability of charging. Are the ambient aerosols spherical?*

A: We assumed that ambient aerosols are spherical in this study. We added sentence of this assumption in the text (Line 201-202).

"As in Section 4.1, the measured aerosol number concentration data were fitted to the log-normal distribution for calculation and the aerosols are assumed to be spherical"

*Q2: Was there any contribution from other physical factors (e.g., mixing state surface activity, viscosity, non-spherical morphology) on the uncertainties in? Did the authors take any measures to control the contribution from the aforementioned and other sources of uncertainty in?*

A: As reviewer commented, CCN activation is affected by combined physical factors including mixing state, surface activity, viscosity and particle morphology. Also, these physical factors themselves can influence the CCN activation by multiply charged particles. In other words, other physical factors can contribute uncertainties of CCN activation. For example, particle morphology can influence in determining the diameter corresponding to the centroid mobility of each size bin. Altaf (2018) explored the effect of size-dependent morphology on CCN activity using aerosol particles composed of organic compounds and salts. The difference in activation can be explained by a homogeneous and phase-separated morphology. In this study, we focus on the effect of multiply charged particles on CCN activation. For laboratory experiment, we consider the particle morphology by applying the particle shape factor when calculating κ values. For examining the uncertainties of activation fraction by morphology, we calculate relative uncertainties (=absolute error / measured value). It does not exceed 1% when applying the shape factor of

ammonium sulfate (χ=1.02) and sodium chloride (χ=1.08). Unlike calibration experiment, other physical factors cannot be controlled during the ambient aerosol measurement. For the BS2-CCN system, aerosol and sheath flow, and supersaturation in the CCNC are fixed during the particle size scan process of the CCN activation curve, uncertainties in CCN activation due to changes in flow rate and supersaturation can be minimized. Figure R1 showed exemplary $D_p$ scan (20-100nm) of lab generated ammonium sulfate. When scanning the $D_p$, absolute deviation of $F_{act}$ was mostly less than 0.05 except the first 10 seconds of each scan and $F_{act}$ value is higher than 0.85. We minimize the uncertainty by defining the first 10 seconds of each $D_p$ scan as a stabilization time and excluding it when calculating the $F_{act}$ value.

[Figure]

**Figure R 1. Exemplary $D_p$ scan (20-100 nm) of lab generated ammonium sulfate. (a) 1 second data of activated fraction ($F_{act}$), marked in black dot (left ordinate), particle diameter (red line, right ordinate). Average and standard deviation of each diameter (30-second average data except for 10 seconds of stabilization) are presented in orange square with bar. (b) Absolute deviation of $F_{act}$ (black dot, left ordinate) and particle diameter (red line, right ordinate): The grey shaded box indicates the stabilization time (~10 seconds) of each particle diameter. (c) Particle number concentration ($N_{CN}$, red dot) and CCN number concentration ($N_{CCN}$, blue dot). Time resolution of each data point is 1 second and the particle diameter is changed every 40 seconds. $S_{max}$ is set to be 10 K (0.8%). Reprinted from Kim et al. (2021) under the Creative Commons Attribution 4.0 License.**

We added the sentences to explain possible uncertainties including combined physical factor that reviewer mentioned (Line 241-251).

"In addition to the effect of multiply charged particles on CCN activation, there are other possible

uncertainties of CCN activation in the BS2-CCN system. It can be affected by combined physical factors including mixing state, surface activity, viscosity and particle morphology. For example, CCN number concentration can be underestimated in the ultrafine mode with a high organic mass fraction due to the lower water surface tension. Particle morphology can affect CCN activity as it can influence in determining the diameter corresponding to the centroid mobility of each size bin. Altaf et al. (2018) explored the effect of size-dependent morphology on CCN activity using aerosol particles composed of organic compounds and salts. For calibration experiment, particle morphology is considered by applying the particle shape factor when deriving κ values. The relative uncertainties, the ratio between absolute error and measured value, do not exceed 1% for calibration aerosols. However, the relative uncertainties could be higher in ambient aerosols due to the non-sphericity. For BS2-CCN system, the absolute deviation of $F_{act}$ was mostly less than 0.05 except for the stabilization time, the first 10 seconds for each $D_p$ scan and the $F_{act}$ value that is higher than 0.85 (Kim et al. 2021)."

*Q3: Page 4 – The authors did a good job of describing the formulation of the new kernel function for multiple charge correction in the CN and CCN number concentrations measured using the DMA 3080. Traditionally, the charge correction for the CN and CCN measurements from the DMA 3080 is done using the Weidensohler (1988) method in the SMCA. Was any significant difference observed between the number concentrations obtained from the 2 charge correction methods? On Line 79, Is charging theory - Wiedensohler 1988 applied? Some clarification on this part of the text could elucidate differences and similarities in theories applied.*

A: In this study, we adopted methods of Wiedensohler (1988) and, Gunn and Woessner (1956) for charge correction. The Kernel function is the same with traditional methods but the main difference of our method is calculation of CCN number concentration inside the algorithm. Here, we calculated CCN number concentration by doubly/triply charged particles using the $F_{act}$ function, $h(x, \varphi, D_p^*)$. we added more detailed description in the algorithm part in section 2.3 and sentence in the text to describe charge correction (Line 101-102).

"The particle charge distribution at each size can be calculated according to the Wiedensohler (1988) and, Gunn and Woessner (1956)."

*Q4: How much is the overall improvement in the size-resolved activation ratio of the aerosol compared to the traditional SMCA approach and is this difference statistically significant?*

A: Our method cannot be directly compared to a traditional SMCA. The reasons are presented in the introduction part: 1) BS2-CCN system uses multiple and continuous supersaturations in the chamber, not a single supersaturation used in commercial DMT-CCNC. 2) The size-resolved $F_{act}$ value is directly used to derive κ value, whereas traditional methods including SMCA aim at finding a critical diameter or supersaturation. The improvement in the size-resolved activation ratio by algorithm is shown in Fig. 7 which presents the deviation (original value – corrected value) $F_{act}$ of two cases of ambient aerosols.

*Q5: Page 6 line 160 – The authors mention that calibration results obtained using the charge correction algorithm may be closely replicated with a minimized influence of multiply charged particles, if*

*"experimental control is performed well". What does that mean? Were the experimental conditions varied across different calibration procedures? What type of experimental control would be required to obtain high quality calibration without the use of charge correction algorithms?*

A: Kim et al. (2021) examined the effect of multiply charged aerosols on calibration curve through CCN activation model and experiments. The CCN activation model describes the CCNC response to the transferred polydisperse charge-equilibrated particles through an ideal DMA and calculates an idealized CCN instruments response with an assumed lognormal particle size distribution. The $F_{act}$ by doubly charged particles not only increases as the value of $D_g$ increases, but also $\sigma_g$ increases as the increases even if the is the same (Fig. R1). Therefore, when generating calibration aerosols, small $D_g$ and $\sigma_g$ in number size distribution are recommended to minimize the effect of multiply charged particles on the calibration curve. These effects can also be seen in the calibration experiment using sodium chloride. Both calculation and experiment results imply that the number size distribution of the generated particles should be considered and it is recommended to generate aerosols with $D_{peak}$ corresponding to an $F_{act}$ less than 0.5. The number size distribution of generated calibration aerosols can be controlled by adjusting the particle concentration. We added the sentence to clarify experimental control part (Line 185-187).

"Kim et al. (2021) examined the effect of multiply charged aerosols on calibration curve through CCN activation model and experiments. By adjusting the particle number concentrations, number size distribution with a small $D_g$ and $\sigma_g$ could be generated to minimize the influence of multiply charged particles."

[Figure]

Figure R 2. Calculated ideal activation fraction for log-normally distributed, charge-equilibrated particles transmitted BS2-CCNC system. Shown are (a) assumed log-normal particle size distribution (black solid line, left ordinate, $N = 2000\ cm^{-3}, D_g = 50\ nm, and\ \sigma_g = 1.5$ ), total activation fraction (red solid line), activation fractions by singly charged particles (red dashed line) , (b) activation fraction by singly charged particle (red solid line) and doubly charged particles (red dashed line), and the ratio of [+2]/[+1] charges (blue solid line), which refers to $f(D, n = +2\ )/f(D, n = +1)$ with mobility diameter at charge equilibrium. $f(D, n)$ is the fraction of particle carrying $n$ charges at charge equilibrium by Wiedensohler (1988) and (c) activation fractions by doubly charged particles ($F_{act\_double}$) for variant particle size distributions. Information of each particle size distribution is presented in the legend of the figure. Reprinted from Kim et al. (2021) under the Creative Commons Attribution 4.0 License.

Q6: *The authors mention that the hygroscopicity parameter was derived from the formulation given by Petters and Kreidenweis (2007). This suggests that the uncertainties in the at the point of activation (which result from there being multiply charged particles in the population corresponding to the dry activation ratio) will directly relate to the uncertainties in. Moreover, the uncertainties due to different multiply charged particles will likely have different magnitude. How do these uncertainties in the size-resolved activation ratio translate to the uncertainties in the aerosol? Furthermore, is there any correlation between the uncertainties due to specific multiply charge particles and the charge that they*

*carry?*

A: Uncertainties of size-resolved activation ratio connected to κ can be explained by Fig.R3 and Fig. R4. Figure R3 presents the κ distribution which corresponds to the $F_{act}$ values of a particle ranged from 50 nm to 150 nm. Figure R4 shows the distribution of the absolute deviation of κ when there is a 0.05 difference from the original $F_{act}$ value. As the $D_p$ increases and the $F_{act}$ value increases, the deviation of κ becomes larger.

And, it can be sized that there is a correlation between the uncertainties caused by specific multiply charged particles and the charge that they carry. The uncertainty by particle morphology could be an example. As mentioned above, particle morphology could affect the determination of particle size in DMA and particle diameter corresponding to the doubly and triply charges can be affected accordingly. Depending on the particle number size distribution, the activation fraction induced by multiply charged particles can be different. These are described in the Fig.2 and Section 4.

[Figure]

**Figure R 3. κ distribution which corresponds to the $F_{act}$ values of a particle ranged from 50 nm to 150 nm. It is calculated from the fitting curve of $F_{act} - S_{aerosol}$ relation for dT=8 K condition. Reprinted from Kim et al. (2021) under the Creative Commons Attribution 4.0 License.**

[Figure]

**Figure R 4. Distribution of absolute deviation of κ when there is a 0.05 difference from the original $F_{act}$ value. The same fitting curve of of $F_{act} - S_{aerosol}$ relation for dT=8 K condition used in Fig.R3 is applied for the calculation.**

*Q7: For the test on the ambient aerosols – What was the chemical composition of the ambient aerosols which were analyzed using the new algorithm? The quantified uncertainties in was helpful, however did the authors verify what proportion of these uncertainties in were due to the multiple charging problem?*

A: We measured the mass concentration of PM1 chemical composition (organics, sulfate, nitrate, ammonium and chloride) using aerosol chemical speciation monitor (ACSM). For Case I, organics occupied about 50.4% among the composition followed by sulfate (26.8%), ammonium (12.3%), nitrate (10.0%) and chloride (0.5%) whereas sulfate accounted for the highest percentage (43.0%) followed by organics (33.0%), ammonium (18.0%), nitrate (5.0%) and chloride (1.0%). For quantifying the proportion of uncertainties by possible factors including multiply charged problem, we need additional experiments. Especially, size-resolved chemical composition data which is not available in this study is necessary. Instead, we add the description of possible uncertainties and calculate the relative uncertainty of CCN activation by the particle morphology which can also affect the CCN activation by multiply charged particles.

*Q8: Figure 2(b) is confusing. What are the sizes of the particles that carrying doubly and triply charges? Are they the particles with the same mobility of the singly charged particles, or the probability of the size of the particles being doubly or triply charged? Are the doubly and triply charged particle sizes in the 95% of the Gaussian distribution? It is suggested to mention that fraction of doubly and triply charged particles depends on the number concentration of the larger particles.*

A: Thanks for your suggestion. Particle number fraction at each charge in Fig. 2b is calculated with Kernel function (Fig. 1) and particle number size distribution. Specifically, number fractions of singly and doubly

charged particles are 0.88 and 0.11, respectively, when particle diameter (i.e., DMA set size) is set to be 70.6 nm (Fig.2b). According to Fig.1, peak diameters of Kernel function for double and triple charging particle are about 103.8 nm 131.4 nm that are within the 95% of the confidence interval when DMA set size is 70 nm. It is noted that particle number size distribution is assumed to be log-normal distribution. As you suggested, we added the sentence in the revised manuscript (Line 153-154)

"And, the number fraction of doubly and triply charged particles depends on the number concentration of the larger particles"

---

## Author Comment (AC2)

**Response to Reviewer #3**

This article presents a new algorithm for processing measurements from a Broad Supersaturation Scanning Cloud Condensation Nuclei (BS2) system. The outcome of this algorithm is to improve retrieval of aerosol hygroscopicity parameters, namely the particle hygroscopicity, $\kappa$. The article claims that this algorithm provides a unique solution to a known problem in determining $\kappa$: namely, that multiply-charged particles that pass through the differential mobility analyzer (DMA) in a BS2 system result in misshapen particle activation curves which degrade the retrieval of $\kappa$. Despite the claim of novelty, the algorithm bears a rather close resemblance to the proposed methodology of Moore et al (2010).

A: We thank the reviewer for sincere and helpful comments on our manuscript. We did our best to answer and correct what the reviewer pointed out. We believe that the quality of our manuscript is improved as we reflect the reviewer's comments. Below each of the summarized questions/comments is written with the Italic font and then our response is followed with the normal font.

Comments

*Q1: The Methods section is missing sufficient detail for their method to be utilized and reproduced by other researchers. Some issues are purely technical: The authors need to re-work the notation of the Method section. There are several instances where the notation is not appropriate, misleading, or definitions are missing altogether. Other issues are pragmatic: further description of their BS2-system should be included (rather than referenced) such as impactor size, DMA size detection range, etc. This is essentially, a methods paper and the Methods section is perhaps the weakest point of the current manuscript. It should be spelled out the letter what a researcher needs to do to implement this method.*

A: As the reviewer suggested, we modified the method part by adding details. Each detailed question (in supplement file) and answer are written below.

*Q1.1: The use of "$\upsilon$" as the number of elementary charges is unusual and confusing. The variable "$\upsilon$" ought to be reserved for kinematic viscosity in this context. Some authors use $\varphi$ to represent the number of charges, e.g. Collins, Flagan & Seinfeld (2003).*

A: We changed "$\upsilon$" to "$\varphi$" for the number of elementary charges in Section 2.3.

*Q1.2: The authors omit the definition of the set mobility, $Z_p^*$, which is not at all equivalent to $Z_p$. This might not be obvious to a reader at first pass, so $Z_p^*$ must be defined.*

A: $Z_p$ is an electrical mobility, which can be calculated by Eq. (1) and $Z_p^*$ is set electrical mobility. We added the definition in the revised manuscript (Line 100).

*Q1.3: It may be more correct to say that Wiedensohler (1986) developed an empirical model based on charging theory of Fuchs (1963). Additionally, Wiedensohler's model is only valid up to ±2 charges. Beyond this number of charges, it is common to use Gunn (1956). As you evaluated charging probability for +3 charges, you would need to apply the formulae described by Gunn.*

A: In this study, we adopted the method of Wiedensohler (1988) and, Gunn and Woessner (1956) to calculate the particle charge distribution. We added the sentence for calculation of the particle charge distribution in the revised manuscript (Line 101).

"The particle charge distribution at each size can be calculated according to the Wiedensohler (1988) and, Gunn and Woessner (1956)."

*Q1.4: To avoid confusion, $G_v$ should be redefined:*

$$G_v(D_p, x) = F(x, v)\Omega(x, v, D_p)$$

*Without the summand. It should also be specified how $\Omega(x, v, D_p)$ was calculated.*

A: As the reviewer suggested, we redefine $G_v$ (Eq. (3)). And, $\Omega(x, \varphi, D_p)$ represents the probability of particle that pass through the DMA. It can be calculated by the transfer function for a cylindrical DMA column. The transfer function is a piecewise linear probability function of triangular shape and can also be presented using electrical mobility, $\Omega(Z)$. Specifically, $\Omega(Z_p^*) = 1$, $\Omega(Z_p^* - 0.5\Delta Z_p) = \Omega(Z_p^* + 0.5\Delta Z_p)$, and $\Omega(Z_p^* - \Delta Z_p) = \Omega(Z_p^* + \Delta Z_p) = 0$ (Kim et al. 2021). We added the sentence to describe $\Omega(x, \varphi, D_p)$ (Line106-107).

"$\Omega(x, \varphi, D_p^*)$ is the probability of particles of $D_p^*$ that pass through the DMA, which is a piecewise linear probability function of triangular shape."

*Q1.5: further description of their BS2-system should be included (rather than referenced) such as impactor size, DMA size detection range, etc.*

A: In the beginning of Section 2, we added the detailed description of BS2-CCN system with the figures of schematic plot of the BS2-CCN system and a newly designed inlet (Fig.R1 and Fig.R2) which is adopted from Kim et al. (2021) (Line 86 - 93).

[Figure]

**Figure R 1. Schematics of typical CCN ((a), (c) and (e)) and BS2-CCN measurement ((b), (d) and (f)). (a) and (b) Contour of supersaturation in the CCN activation unit and configuration of aerosol and sheath flow; (c) and (d) Distribution of supersaturation in the activation unit ($S_{tube}$). $r$ is the radial distance to the centerline. The shaded areas represent the sheath flow part, and non-shaded areas represent the aerosol flow part. (e) and (f) Plot of the activation supersaturation of aerosol particles $S_{tube}$ against the activation fraction $F_{act}$. Reprinted from Su et al., (2016) under the Creative Commons Attribution 4.0 License.**

[Figure]

**Figure R 2. (a) Front perspective view of an embodiment of the diffusive inlet. (b) Longitudinal sectional (the cross section in X-Z surface) view of Fig.A1 (a).   Each of numbers in the figure is as follows:   main body (101), a sheath flow straightener (102), an aerosol inlet (103), a funnel-shaped region (105) where the cross-section of the aerosol is smoothly expanded, the angle (106) of the wall of the funnel-shaped region, the inlet of sheath air (104) at the side of the main body and two rubber O-rings (107) at the lower end of the main body to keep the activation tube air-tight. Reprinted from Kim et al., (2021) under the Creative Commons Attribution 4.0 License.**

"The BS2-CCN system contains a modified DMT-CCNC with a newly designed inlet system to measure the size-resolved CCN activity with a high-time resolution. The BS2-CCN system includes a differential mobility analyser (DMA), a condensation particle counter (CPC) and a modified DMT-CCNC. Figure S1 and S2 present the schematic plot of a BS2-CCN system and a newly designed inlet, respectively, which are adopted from Su et al. (2016) and Kim et al. (2021). Selected monodisperse particles by DMA enter into CPC and a modified CCNC, respectively and thereby the size-resolved $F_{act}$ values can be obtained. A new inlet system of a modified CCNC makes a stable low sheath-to-aerosol flow ratios (SAR), which obtain a monotonic $F_{act} - S_{aerosol}$ relation. The aerosol and sheath flow for a modified CCNC are set to 0.46 L min$^{-1}$ and 0.04 L min$^{-1}$, respectively. A detailed description of BS2-CCN system is described in Kim et al. (2021)."

Q2: The results demonstrate a fulfillment of the original promise. The appearance of multiply charged particles in the activation curve have disappeared. However, the assessment of this methodology is fairly qualitative. The case study approach is not sufficient enough argument for researchers to understand when this correction needs to be applied to their measurements. It is clear, for example, that the correction algorithm need not to be applied to calibration experiments. A revision of this manuscript should include more quantitative laboratory-based study with ammonium sulfate rather than the qualitative field-base study that is currently used.

A: As reviewer suggested, we added additional calibration results of ammonium sulfate and sodium chloride in the supplementary material (Fig.S3 and Fig.S4) in the revised manuscript (Here in Fig.R3 and R4). We combined experiment results of ammonium sulfate in the Fig. R3 which indicates activation fraction ($F_{act}$) curves for dT = 6, 8 and 10 K. Square and error bar indicate the average and standard deviation of measurement points, respectively. Like Fig.3 in the revised manuscript, small plateau disappears and the other $F_{act}$ value slightly decrease. Additionally, we added the calibration result of sodium chloride in Fig.S4 (Here in Fig.R4) which shows the consistent result with Fig. 3 and Fig.S4.

We revised and added the sentences for additional calibration results (Line 187 – 190).

"Additionally, Fig. S3 and S4 show the combined result of ammonium sulfate calibration experiments for three dT conditions and sodium chloride experiment, another representative calibration aerosol. All show consistent results with the Fig.3 indicating that the suggested algorithm performs well."

[Figure]

**Figure R 3. Activation fraction ($F_{act}$) curve for dT = 6K (red), 8K (green) and 10K (orange). Square and error bar indicate the average and standard deviation of measurement points, respectively. Solid and dashed line with squares indicate results before and after the correction.**

[Figure]

**Figure R 4. Particle (red solid line) and CCN (red dashed line) number size distribution, and activation fraction ($F_{act}$) curve before (black solid line) and after (black dashed line) correction for dT = 6 K of sodium chloride (NaCl) particles. Squares indicate measurement points.**

Q3: The revision should also include a full uncertainty analysis to determine confidence intervals on derived hygroscopicity. This would allow other researchers to better understand when they should apply this correction and the magnitude of the effect on hygroscopicity retrieval (so that they can troubleshoot their implementation). The authors should also make it more apparent in the abstract and conclusions that the proposed algorithm assumes that the particle size distribution is monomodal. The algorithm has not been tested for more complicated PSDs.

A: For hygroscopicity retrieval, we revised the Fig.6 (Here, Fig.R6) in the revised manuscript including the uncertainties (shaded area). The uncertainties of hygroscopicity are evaluated from $F_{act}$ value. The CCN activation is affected by combined physical factors including mixing state, surface activity, viscosity and particle morphology. For example, particle morphology can influence in determining the diameter corresponding to the centroid mobility. For laboratory experiment, we consider the particle morphology by applying the particle shape factor when calculating κ values. For examining the uncertainties of activation fraction by morphology, we calculate relative uncertainties (=absolute error/measured value). It does not exceed 1% when applying the shape factor of ammonium sulfate (χ=1.02) and sodium chloride (χ=1.08). For the BS2-CCN system, aerosol and sheath flow, and supersaturation in the CCNC are fixed during the particle size scan process of the CCN activation curve, uncertainties in CCN activation due to changes in flow rate and supersaturation can be minimized. Figure R5 showed exemplary $D_p$ scan (20-100nm) of lab generated ammonium sulfate. When scanning the $D_p$, absolute deviation of $F_{act}$ was mostly less than 0.05 except the first 10 seconds of each scan and $F_{act}$ value is higher than 0.85. We minimize the uncertainty by defining the first 10 seconds of each $D_p$ scan as a stabilization time and excluding it when calculating the $F_{act}$ value.

[Figure]

**Figure R 5. Exemplary $D_p$ scan (20-100 nm) of lab generated ammonium sulfate. (a) 1 second data of activated fraction ($F_{act}$), marked in black dot (left ordinate), particle diameter (red line, right ordinate). Average and standard deviation of each diameter (30-second average data except for 10 seconds of stabilization) are presented in orange square with bar. (b) Absolute deviation of $F_{act}$ (black dot, left ordinate) and particle diameter (red line, right ordinate): The grey shaded box indicates the stabilization time (~10 seconds) of each particle diameter. (c) Particle number concentration ($N_{CN}$, red dot) and CCN number concentration ($N_{CCN}$, blue dot). Time resolution of each data point is 1 second and the particle diameter is changed every 40 seconds. $S_{max}$ is set to be 10 K (0.8%). Reprinted from Kim et al. (2021) under the Creative Commons Attribution 4.0 License.**

We added the figure (Fig.6 in the revised manuscript) of hygroscopicity retrieval with uncertainties and sentences to explain possible uncertainties (Line 241-251). Also, we mentioned about monomodal distribution in the abstract.

"In addition to the effect of multiply charged particles on CCN activation, there are other possible uncertainties of CCN activation in the BS2-CCN system. It can be affected by combined physical factors including mixing state, surface activity, viscosity and particle morphology. For example, CCN number concentration can be underestimated in the ultrafine mode with a high organic mass fraction due to the lower water surface tension. Particle morphology can affect CCN activity as it can influence in determining the diameter corresponding to the centroid mobility of each size bin. Altaf et al. (2018) explored the effect of size-dependent morphology on CCN activity using aerosol particles composed of organic compounds and salts. For calibration experiment, particle morphology is considered by applying the particle shape factor when deriving κ values. The relative uncertainties, the ratio between absolute error and measured

value, do not exceed 1% for calibration aerosols. However, the relative uncertainties could be higher in ambient aerosols due to the non-sphericity. For BS2-CCN system, the absolute deviation of $F_{act}$ was mostly less than 0.05 except for the stabilization time, the first 10 seconds for each $D_p$ scan and the $F_{act}$ value that is higher than 0.85 (Kim et al. 2021).

[Figure]

**Figure R 6. κ values as function of particle size before (filled square) and after (square) correction for Case I (black) and Case II (red). Shaded area indicates uncertainties of κ values which is evaluated from the $F_{act}$ values.**

Q4: The authors should support their claim that their methodology is a necessary improvement on previous approaches. A revision of this manuscript should also include a side-by-side comparison of this methodology to previous proposed methods in the literature, e.g., Moore, Nenes & Median (2010) to which the proposed method bears an uncanny resemblance.

A: For the BS2-CCN system, multiple and continuous supersaturation in the chamber, not a single supersaturation used in the commercial DMT-CCNC. It is the significant difference between the BS2-CCN system and the existing CCNC. Particularly, CCN number concentration can be calculated as follows:

$$N_{CCN}(D_{Z*}) = \sum_{n=1}^{3}\left[\int_{Z\,=\,Z*\,+\,\Delta Z}^{Z\,=\,Z*\,-\,\Delta Z} h(D_{Z,n})f(D_{Z,n},n)\Lambda(Z)\frac{dN_n}{dZ}dZ\right] \tag{R1}$$

where $h(D_{Z,n})$ is a function for a fraction of particles that activate as cloud droplets. For the BS2-CCN system, the activation fraction, $h(D_{Z,n})$, is calculated by Eq. (S2) and (S3). $dN_n/dZ$ is the differential size distribution of +n-charged particles in the mobility domain. It is noted that electrical mobility $Z^*$ is based on a particle with +1 charge $(D_{Z*,1})$, which should be assumed to define $\Lambda(Z)$. The activation fraction of aerosol particles with the same $S_{aerosol}(D_{Z,n})$ is calculated by integrating the activation fraction function $g(x)$ and flow velocity $v(r)$ over the cross-section of the aerosol flow.

$$h(D_{Z,n}) = \frac{2\pi \int_0^r vg(S_{aerosol}(D_{Z,n}) - S_{tube}(r))r dr}{2\pi \int_0^r vr dr} \tag{R2}$$

$$g(x) = \begin{cases} 1 & if \ x \leq 0 \\ 0 & if \ x > 0 \end{cases} \tag{R3}$$

here $r$ is the radial distance to the centerline of the activation unit (i.e., $r = 0$ for the center). $S_{tube}(r)$ is a typical distribution of supersaturation in the activation tube of CCNC and can be calculated as $S_{tube}(r) = S_{max} \times cos(0.14 \times r)$.

Here, we use the observed activated fraction to calculate the CCN number concentration by doubly/triply charged particles similar to the method of Moore et al. (2010). And updated activated fraction is calculated by subtracting the CCN number concentrations by multiply charged particles and is directly used to derive hygroscopicity parameter rather than iteration to derived critical diameter of activation curve. We describe other traditional methods including Moore et al. (2010) and reasons why we develop the method for the BS2-CCN system, especially for continuous and multiple supersaturation distribution in the chamber as well as differences in the introduction part (Line 47-76).

*Q5: Add the importance of obtaining the hygroscopicity parameter, kappa, into the introduction part*

A: As suggested, we added the importance of obtaining the hygroscopicity parameter, κ, into the introduction part (Line 55-59)

"Ultimately, these methods are to find the true critical diameter or supersaturation that we want to obtain to derive the κ value. The single hygroscopicity parameter, κ, is used to model the composition-dependence of the solution water activity. It can be used as a proxy for the chemical composition model and thereby streamline aerosol composition model. Also, the values can manage the hygroscopic properties of complex aerosol types."

*Q6: What was the value of geometric mean diameter and geometric standard deviation? Even if the reader can go read your other paper, re iterate the suggested limits of Dg and sigma g for a "well-performed" calibration experiment here. Finally, Table 1 could probably be added to Supplemental Information documents, I don't think there is any benefit to it appearing in the main text.*

A: Kim et al. (2021) examined the effect of multiply charged aerosols on calibration curve through CCN activation model and experiments. The CCN activation model describes the CCNC response to the transferred polydisperse charge-equilibrated particles through an ideal DMA and calculates an idealized CCN instruments response with an assumed lognormal particle size distribution. The $F_{act}$ by doubly charged particles not only increases as the value of $D_g$ increases, but also $\sigma_g$ increases as the increases even if the is the same (Fig. R4). Therefore, when generating calibration aerosols, small $D_g$ and $\sigma_g$ in number size distribution are recommended to minimize the effect of multiply charged particles on the calibration curve. These effects can also be seen in the calibration experiment using sodium chloride. Both calculation and experiment results imply that the number size distribution of the generated particles should

be considered and it is recommended to generate aerosols with $D_{peak}$ corresponding to an $F_{act}$ less than 0.5. The number size distribution of generated calibration aerosols can be controlled by adjusting the particle concentration. We added the sentence to clarify experimental control part (Line 185-187). Also, we moved Table 1 from the revised manuscript to supplemental information document.

"Kim et al. (2021) examined the effect of multiply charged aerosols on calibration curve through CCN activation model and experiments. By adjusting the particle number concentrations, number size distribution with a small $D_g$ and $\sigma_g$ could be generated to minimize the influence of multiply charged particles."

[Figure]

**Figure R 7. Calculated ideal activation fraction for log-normally distributed, charge-equilibrated particles transmitted BS2-CCNC system. Shown are (a) assumed log-normal particle size distribution (black solid line, left ordinate, $N = 2000\ cm^{-3}, D_g = 50\ nm, and\ \sigma_g = 1.5$ ), total activation fraction (red solid line), activation fractions by singly charged particles (red dashed line) , (b) activation fraction by singly charged particle (red solid line) and doubly charged particles (red dashed line), and the ratio of [+2]/[+1] charges (blue solid line), which refers to $f(D, n = +2 )/f(D, n = +1)$ with mobility diameter at charge equilibrium. $f(D, n)$ is the fraction of particle carrying $n$ charges at charge equilibrium by Wiedensohler (1988) and (c) activation fractions by doubly charged particles ($F_{act\_double}$) for variant particle size distributions. Information of each particle size distribution is presented in the legend of the figure. Reprinted from Kim et al. (2021) under the Creative Commons Attribution 4.0 License.**

Q7: It should be noted that ambient aerosol size distributions are rarely monomodal, like this retrieval assumes. The authors should thus comment on what environments, if any, it is safe to assume that the

distribution is monomodal. In the marine environment, the particle size distribution is often at least tri-modal.

A: Thank you for your comment. We agree that bimodal and/or trimodal aerosol distributions which the algorithm doesn't consider can be observed along with monomodal distribution in the ambient environments. The graph below (Fig.R2) presents the aerosol size distribution during the first period of ship campaign. During the first three days, ship sailed to the South (31.5°N, 125.0°E), which was relatively less affected by the continent. And then, ship crossed the north and south of the Yellow Sea alternately. Also, there was a period for the transportation of Asian dust (March 28-29). As shown in Fig.R2, both monomodal and bimodal aerosol distribution were observed during the measurement period. Therefore, we chose two cases which are monomodal aerosol distributions for the algorithm application: One from the period of voyage south and one from the period of Asian dust. As reviewer suggested, we add sentence about aerosol size distribution in the ambient environments (Line 217-219).

"However, it is noted that not only monomodal distribution but also bimodal/trimodal aerosol distribution are observed in the ambient environments, and care should be taken to apply the correction algorithm accordingly."

[Figure]

Figure R 8. Aerosol number size distribution measured from SMPS during the first period of ship campaign (March 25 – April 2).

Q8: The referral of the reader to another paper for information about your campaign is frustrating. Basic information like the duration of the campaign should be specified. The selection of cases should also be detailed and better motivated. It is clear after reading this section that you want to highlight the degree to which $D_g$ determines the magnitude of the effect your multiple-charge correction will have, but this should be spelled out at the beginning of the section.

A: We added the detailed description of the ship campaign (Line 194-200).

"Here, we adopted cases from the polluted marine aerosol measurement that was held in the Yellow Sea during springtime, 2021. Specifically, the campaign was held three times, about 10 days each (1st period: 22 March – 2 April, 2nd period: 6 – 15 April and 3rd period: 20 – 29 April), and crossed the north and south of the Yellow Sea alternately. During the first observation period, ship went down to the South (31.5°N, 125°E), which was relatively less affected by the continent. Also, there was a period for the transportation of Asian dust. Among the first observation period, we chose two cases with different aerosol size

distributions; one from the period when the ship sailed south and one from the period when Asian dust was transported."

Q9: I also believe that this effect could have been better studied directly in the lab with particles of known hygroscopicity, e.g. ammonium sulfate, by modifying the atomizer pressure & solution concentration. This would allow you to directly relate the deviation in κ as a function of $D_g$, Uncertainty analysis could then allow you to determine a threshold $D_g$ below and above which your correction should/shouldn't applied. This would provide your readers with a much more satisfying quantitative assessment than the relatively qualitative case study that is presented.

A: According to Fig. R2 (c), the CCN activation by multiply charged particles is affected not only by $D_g$, but also by $\sigma_g$. So, it is difficult to suggest a $D_g$ threshold directly whether the correction algorithm should be applied or not. This can also be seen in Fig. 7 in the revised manuscript. Figure 7 presents the deviation of $F_{act}$ value as function of $D_g$ for two different cases. The patterns of $F_{act}$ and κ deviation are quite different between Case I and Case II due to the different.

Q10: Next, $F_{act}$ is a function of particle size. Are you saying that the average of $F_{act}$ decreased from 0.01 to 0.07? Or, are you referring to a specific value of $F_{act}$ at a diameter of 60 nm? Follow up question: How can something decrease from 0.01 to 0.07?

A: No, it means the range of decrease for $F_{act}$ value when applying the correction algorithm. And the particle diameter with the largest reduction of $F_{act}$ value was around 70 nm, near the geometric mean diameter of the aerosol size distribution. For avoiding the confusion, we rewrote the sentence (Line 203-205).

"When applying the algorithm, the reduction in the Fact value of the activation curve is 0.01 to 0.07. In particular, the decrease was the largest at about 70 nm, near the geometric mean diameter of the aerosol size distribution."

Q11: I would like to see a proper error analysis added to this section. It is clear that your algorithm results in changes to retrieved values of hygroscopicity, κ. It also seems to complete the task as advertised, removing multiply charged particles from the minor plateau, but are these changes significant? Add 95% C.I. to your estimates of κ in Fig.6. This will help quantitatively reinforce your claim that multiple-charge corrections need only be considered when the peak of the observed size distribution is > 100 nm.

A: Some existing correction methods pointed out that a minor plateau of the activation curve induced by multiply charged particles can be ignored as the main goal of correction methods is to find the activation diameter to derive κ values. However, for the BS2-CCN system, the activation fraction ($F_{act}$) value is directly connected to κ values. In other words, the minor plateau could be significant if we are interested in κ value of corresponding particle diameter. We added uncertainties of derived κ values in Fig. 6 in the revised manuscript (Here in Fig.R6).

Q12: Uncertainty/Error analysis should also inform the discussion of Fig.7. This would help support your

claim that these deviations are meaningful by demonstrating that they are statistically significant.

A: As described in the Section 4, $F_{act}$ values by multiply charged particles can be affected not only by $D_g$ but also by $\sigma_g$ which can explain how broad the particle number size distribution is. It is also related to particle morphology. Particle morphology could affect the determination of particle size in DMA and particle diameter corresponding to the doubly and triply charges can be affected accordingly. Depending on the particle number size distribution, the activation fraction induced by multiply charged particles can be different. We added the description of uncertainty in the Section 4.

Q13: As it currently stands, you have demonstrated that your correction algorithm works for monomodal aerosol. It is a rather large stretch that you finish this manuscript by saying it should be applied to "a variety of particle number size distributions." Remove this statement or prove it.

A: We remove the statement of "a variety of particle number size distribution" as the reviewer commented as we only discuss monomodal particle distribution in this study.

**Technical Corrections**

*Line 28: "the **a** key element". I disagree with the notion that clouds are the sole element controlling climate change.*

A: Corrected.

*Line 29: Change to "Despite the scientific importance of CCN,"*

A: Corrected.

*Line 30: "aerosol-cloud interaction**s**". There are many types of aerosol-cloud interactions.*

A: Corrected.

*Line 31: Citation needed at the end of "composition and processes." Perhaps the updated version of this figure: https://archive.ipcc.ch/publications_and_data/ar4/wg1/en/tssts-2-5.html.*

A: We added the reference at the end of "composition and processes".

*Line 33: "over the past" → "in recent"*

A: Corrected.

Line 40: "under the simple assumption" Are you referring to a specific assumption here? ZSR? Or, did you mean to say "under **this** simple assumption."

A: "under the simple assumption" used in the Line40 refers to ZSR assumption.

Line 51: On first introduction, explain what Dc is.

A: We add short description of Dc (Line 51)

"The activation efficiency of the particle distribution with the best estimate of critical diameter (Dc), diameter to be activated, is determined by minimizing the statistic by varying the assumed Dc."

Line 52: "Constant fraction by **of** doubly charged particle**s"**

A: Corrected.

Lines 54: Not sure if it is worth mentioning that the process starts from the largest aerosol size bin and iterates towards smaller bins.

A: The part in Line 47 – 55 presents existing correction methods for multiply charged particles. These methods focus on finding the true critical diameter or supersaturation, which is different with the method suggested in this study. Therefore, we think that an additional description of method by Moore et al. (2010) is not necessary in this part.

Line 55: Suggested rephrasing: "Ultimately, each of the methods introduced above are designed to determine the critical activation diameter, Dc, of the test aerosol and thus the hygroscopicity, k, of the aerosol." Additionally, it should be outlined at the beginning of this paragraph that the hygroscopicity parameter, k, is the desired outcome.

A: As suggested, we rephrase the sentence in Line 55 and add description of hygroscopicity parameter, κ, which is the desired outcome.

"Ultimately, each of methods introduced above are designed to determine $D_c$ of the test aerosols and thus the hygroscopicity parameter, κ, of the aerosols. The single hygroscopicity parameter, κ, is used to model the composition-dependence of the solution water activity (reference). It can be used as a proxy for the chemical composition model and thereby streamline aerosol composition model. Also, the values can manage the hygroscopic properties of complex aerosol types."

Line 57: You should introduce the theme of this paragraph in the first sentence. "Whereas previous studies have improved hygroscopicity retrieval through the development of post-processing algorithms, modern studies have focuses on directly manipulating the sampling parameters (e.g. sample flow rate, sheath flow rate, supersaturation, etc.) to allow direct retrieval of k. Examples of this approach include…"

A: Thank you for your suggestion. We added the sentence to introduce the theme of this paragraph (from Line 60) in the first sentence. (Line 60 – 64)

"Whereas previous studies have improved hygroscopicity retrieval through the development of post-processing algorithms, modern studies have focuses on directly manipulating the sampling parameters (e.g. sample flow rate, sheath flow rate, supersaturation, etc.) to allow direct retrieval of κ. Example of this approach include the broad supersaturation scanning (BS2) CCN approach proposed by Su et al. (2016), which modified the inlet and flow system of commercial CCNC to obtain aerosol hygroscopicity and CCN activity with a high time resolution."

Line 58: What is the calibration experiment? Either describe or omit.

A: It means that calibration for the BS2-CCN system. Specifically, the main purpose of calibration is to derive a calibration curve, '$F_{act} - S_{aerosol}$', which is used to derive a hygroscopicity parameter, κ. We revised the sentence (Line 67)

"With the CCN activation model, a response of CCNC with and without considering doubly charged particles was compared and suggested a method for the calibration for the BS2-CCN system to minimize the multiple charge effect."

Line 68: Do you mean continuous as opposed to discrete? Or continuous as in "temporally continuous". It might be beneficial to describe as "continuously variable".

A: For BS2-CCN system, the supersaturation distribution, not a single supersaturation, is made inside the chamber. Therefore, 'continuous' in Line 68 is opposed to 'discrete'. And it is not temporally variable and as well as continuously variable. If the maximum supersaturation is determined in the center of the chamber, a supersaturation distribution is formed inside the chamber due to newly designed inlet system and low SAR. For better understanding, we added Fig.S1 in the revised manuscript which presents the schematic plot of a BS2-CCN system.

Line 82: What is Dp? Be thorough and describe.

A: We added the definition of Dp in Line 98.

Line 84: This equation is valid, but what is $Z^*_p$? $Z^*_p = (Qsh/(2*\pi*V* L))*ln(r2/r1)$

A: '$Z^*_p$' is the set electrical mobility. We added the definition in Line 100.

---

## Author Response (AR2)

**Response to Reviewer #2**

The authors should consider tempering their assertions with respect to calculations of hygroscopicity; as the BS2 may provide excellent data however the interpretations of the data may be flawed.

A: We agreed your comments. In this manuscript, possible uncertainties of the CCN activation including multiply charged particles that can affect to κ calculation are presented. So, we added the sentence (Line 249-251) in the manuscript with respect to calculation of hygroscopicity.

"As κ values can be derived directly by $F_{act}$ values in the BS2-CCN system, these possible uncertainties of CCN activation need to be carefully considered when deriving κ values."

**Response to Reviewer #3**

I commend the authors for having addressed my many previous comments. Readers (and I) can hopefully now appreciate the necessity of this work for a Broad Supersaturation system as the authors have included many useful figures within the supplemental information which outline the key differences between a traditional CCNC and this system. It is useful to have this information available to the reader without having to always refer to previous work. The addition of Subsection 2.1 is welcomed as is the expanded description of your field campaign. Finally, and perhaps best of all, the inclusion of error bars in Figure 6 is extremely useful and demonstrates that there is a statistically significant difference to retrieved k when applying your approach.

There are still some minor technical corrections that would benefit the article.

A: We thank the reviewer for encouraging comments on our manuscript. We also believe that the quality of our manuscript is improved as we reflect the reviewer's comments. Below each of the comments is written with the Italic font and then our response is followed with the normal font.

Minor Comments

*Q1: Line 29: Should read "Despite the scientific importance of CCN, ..."*

A: Corrected (Line 29)

*Q2: Line 61: "...modern studies have focused on ..."*

A: Corrected (Line 61)

*Q3: Lines 97, 144: If you later use $\varphi$ for particle charge (as recommended), it would be good to be consistent here.*

A: Thanks for correcting our mistake. We fixed both (Line 97 and 144)

*Q4: Line 115: "The algorithm assumes the lognormal size distribution for Scanning the size of particles with an Fact of 0 to 1 is performed, and the algorithm applies starting from larger particle to small particle"*

*I think this sentence was meant to be two separate sentences, perhaps:*
*"We note that the algorithm assumes that the particle size distribution is lognormal. Scanning the size of particles..."*

A: We agreed your comment and changed the sentence (Line 115-116).

"We noted that the algorithm assumes the lognormal size distribution. Scanning the size of particles with an $F_{act}$ of 0 to 1 is performed, and the algorithm applies starting from larger particle to small particle."

*Q5: Line 149: This is not a continuous integral over charge. Charge is discrete. Use the appropriate symbol for a discrete summation, Σ.*

A: We changed the discrete summation, Σ of the Eq. (5) (Line 139).

*Q6: Figure 2b: There seems to be some sort of watermark, or white line affecting the readability of 0.8 and 0.2? Could this be fixed?*

A: We change the Fig. 2 clearly (y-axis number) for readability as you pointed out.